# AVROBUSTBENCH: Benchmarking the Robustness of Audio-Visual Recognition Models at Test-Time

**Sarthak Kumar Maharana**    **Saksham Singh Kushwaha**    **Baoming Zhang**[*]
**Adrian Rodriguez**[*]    **Songtao Wei**[*]    **Yapeng Tian**    **Yunhui Guo**
The University of Texas at Dallas, Richardson, TX, USA    [*]denotes equal contribution

## Abstract

While recent audio-visual models have demonstrated impressive performance, their robustness to distributional shifts at test-time remains not fully understood. Existing robustness benchmarks mainly focus on single modalities, making them insufficient for thoroughly assessing the robustness of audio-visual models. Motivated by real-world scenarios where shifts can occur *simultaneously* in both audio and visual modalities, we introduce AVROBUSTBENCH, a comprehensive benchmark designed to evaluate the test-time robustness of audio-visual recognition models. AVROBUSTBENCH comprises four audio-visual benchmark datasets, AUDIOSET-2C, VGGSOUND-2C, KINETICS-2C, and EPICKITCHENS-2C, each incorporating 75 bimodal audio-visual corruptions that are *co-occurring* and *correlated*. Through extensive evaluations, we observe that state-of-the-art supervised and self-supervised audio-visual models exhibit declining robustness as corruption severity increases. Furthermore, online test-time adaptation (TTA) methods, on VGGSOUND-2C and KINETICS-2C, offer minimal improvements in performance under bimodal corruptions. We further propose `AV2C`, a simple TTA approach enabling on-the-fly cross-modal fusion by penalizing high-entropy samples, which achieves improvements on VGGSOUND-2C. We hope that AVROBUSTBENCH will steer the development of more effective and robust audio-visual TTA approaches. Our code is available here.

## 1 Introduction

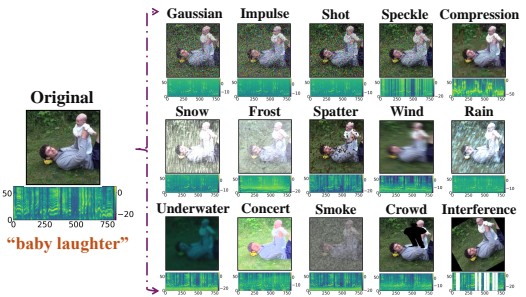

(a) Sample of our 15 proposed corruptions from the "baby laughter" class of VGGSOUND-2C, at a severity level of 5.

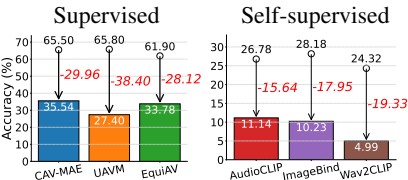

(b) *Significant performance gaps are observed by supervised models [left] and self-supervised models [right]* in terms of mean accuracy, across 15 corruption types at severity level 5 on VGGSOUND-2C, relative to their respective clean performance on VGGSound [10].

Figure 1: AVROBUSTBENCH *comprises diverse and correlated audio-visual corruptions that co-occur in the real world.*

In recent years, the community has seen the rise of audio-visual models [26, 35, 3, 95] that are pre-trained on massive audio-visual data and designed to perform across a wide range of tasks.

39th Conference on Neural Information Processing Systems (NeurIPS 2025) Track on Datasets and Benchmarks.

Table 1: **Benchmark comparison.** AVRobustBench *introduces realistic, co-occurring, and correlated corruptions to audio-visual modalities, in contrast to prior benchmarks. v, a, s and l denote visual, audio, speech and language respectively.* —— indicates an unreported quantity.

| Benchmark | Modalities | Real-World Shifts? | Multimodal Corruptions? | Features | |
|---|---|---|---|---|---|
| | | | | Co-occur? | Correlated? |
| ImageNet-C [38] | {v} | ✓ | ✗ | ✗ | ✗ |
| MULTIBENCH [55] | {a, v} | – | ✗ | ✗ | ✗ |
| YouCook2-P, MSRVTT-P [74] | {l, v} | ✓ | ✗ | ✗ | ✗ |
| SRB [78] | {s} | ✓ | ✗ | ✗ | ✗ |
| Chen et al. [11] | {l, v} | ✓ | ✗ | ✗ | ✗ |
| Hong et al. [40] | {s, v} | ✓ | ✓ | ✓ | ✗ |
| READ [97] | {a, v} | ✓ | ✗ | ✗ | ✗ |
| AVRobustBench (Ours) | {a, v} | ✓ | ✓ | ✓ | ✓ |

While these audio-visual models have been effective in the in-distribution tasks they are trained for, ensuring robustness to distributional shifts, which is underexplored, remains a critical priority for any intelligent system deployed in the real world, i.e., at test-time, especially in safety-critical applications. Consider a scenario involving an autonomous vehicle equipped with audio-visual sensors for scene understanding [65, 33, 52]. Distributional shifts can affect both modalities, e.g., adverse weather conditions like rain, snow, or wind. Such unavoidable real-world distributional shifts challenge the ability of audio-visual systems to perceive their environments accurately, raising a major concern.

While robustness to single and multiple modalities has been studied (see Table 1), studies on joint, correlated distributional shifts remain unexplored. In [38, 37, 15], the robustness of models to image corruptions and perturbations has been deeply detailed. In addition, robustness of models to human speech [6, 94, 51, 78], natural language [89, 93], and multimodal data involving visual and text perturbations [74] have been discussed. There have been works involving robustness to adversarial attacks [18, 7, 86], too. To our knowledge, no prior work systematically analyzes the robustness of state-of-the-art audio-visual models to co-occurring audio-visual corruptions at test time.

To this end, we seek to establish a benchmark for comprehensively analyzing the robustness of open-source state-of-the-art (SotA) audio-visual (AV) recognition models to distributional shifts at test-time, to emulate real-world settings. Building on widely used AV datasets, AudioSet [24], VGGSound [10], Kinetics-Sounds [3] and Epic-Kitchens [16], we introduce AVRobustBench, comprising four audio-visual benchmark datasets: AudioSet-2C, VGGSound-2C, Kinetics-2C, and EpicKitchens-2C. Specifically, we introduce 75 AV corruptions (15 corruptions x 5 severities) that *co-occur* and are *correlated* across both modalities, enabling a large-scale assessment of these models' resilience to challenging, realistic shifts. We refer to this setting as 2C (2 *jointly* corrupted modalities). We group our corruptions into three major categories - *Digital*, *Environmental*, and *Human-Related*, where *Digital* includes gaussian, impulse, shot, speckle, and compression. *Environmental* encompasses snow, frost, spatter, wind, rain, underwater, and concert, smoke, crowd, and interference, are clubbed under *Human-Related*, as in Figure 1a. It is worth emphasizing that the unique challenge in AVRobustBench arises from the real-time occurrence of *correlated* corruptions that *simultaneously* affect both the audio and visual modalities. Our idea of a real-world shift reflects the expectation of correlated shifts that can happen in reality, simultaneously affecting both modalities.

Our **first aim** is to study the robustness of models at test-time. And so, our studied models include SotA AV *supervised* models like UAVM [28], CAV-MAE [30], EquiAV [50], TBN [48], and TIM [8] (Tables 4, 5). We also extend our analysis to *self-supervised* models like AudioCLIP [35], ImageBind [26], and Wav2CLIP [95] under such settings to understand their cross-modal associations. While such models are being widely used in different downstream tasks, it is imperative to understand their robustness and behavior to real world shifts, ensuring a check before being deployed for safety-critical tasks [37] (see Figure 1b).

Distributional shifts are inevitable in the real world [85, 60]. As our **second aim**, we study online *test-time model adaptation (TTA)* that offers a learning paradigm where a pre-trained/source model is adapted to *unlabeled test data* arriving sequentially, under a distributional shift. We evaluate several popular online TTA methods [90, 67, 66, 73] and two recent AV TTA approaches, READ [97] and SuMi [34], on VGGSound-2C and Kinetics-2C. While READ does well with TTA involving only a single-modality corruption (Tables 7, 8), we show that its training objective struggles on our proposed benchmark, involving both audio and visual corruptions (Table 6). Specifically,

through these TTA experiments, we shed light on the existing significant performance gap between a pre-trained model's accuracy on clean test data and its performance after adaptation to proposed audio-visual corruptions.

As an attempt to mitigate this, inspired by [66, 97], we propose AV2C, an online AV TTA approach to perform on-the-fly cross-modal fusion where high-entropy samples, that hurt model adaptation at test-time, are penalized. Also, diverse samples, required for entropy minimization, are selected based on the similarity of current predictions and an exponential moving average of past predictions. Our contributions and findings can be summarized as follows:

- We introduce AVRobustBench, a robustness benchmark for audio-visual recognition models at test-time. This includes four benchmark datasets - AudioSet-2C, VGGSound-2C, Kinetics-2C, and EpicKitchens-2C. Inspired by real-world settings, we propose 75 AV corruptions that *co-occur* and *correlated* across both modalities.

- We find that state-of-the-art audio-visual supervised and self-supervised models show poor robustness, worsening with corruption severity (Figure 2). Contrastively trained self-supervised models, in particular, struggle with noisy unseen cross-modal associations, exposing limits in real-world generalization (Table 4).

- We find that entropy-based norm updates in online TTA approaches, on VGGSound-2C and Kinetics-2C, often lead to overfitting [90, 73, 67]. A recent AV TTA method, READ [97], that adapts QKV parameters for cross-modal fusion also degrades, revealing *modality bias* (Figure 4). Another recent work, SuMi [34], that identifies reliable multimodal samples using interquartile range smoothing and a mutual information loss, also proves limited. Our simple yet effective AV2C that performs on-the-fly cross-modal fusion by minimizing entropy over low-entropy reliable samples, achieves large improvements on VGGSound-2C.

## 2 Related Work

**Audio-Visual Recognition.** With the advent of Transformers [88], Perceiver [43] proposes a modality-specific unified architecture, while Data2vec [5] introduces a training scheme that jointly learns speech, vision, and language. Likewise, PolyViT [56] and VATT [2] share model parameters across modalities. Building on these ideas, UAVM [29] presents a unified transformer architecture. MBT [64] restricts cross-modal fusion through latent tokens introduced in the transformer architecture. AV-MAE [25] explores masked modeling of audio and video and proposes a self-supervised pretraining objective. CAV-MAE [30] combines contrastive learning with masked modeling, while MAViL [42] adds inter- and intra-modal contrastive losses via knowledge distillation [39]. EquiAV [50] extends self-supervision for audio-visual contrastive learning by introducing equivariance. In the works mentioned above, modality-specific transformers are either pre-trained from scratch or initialized with pre-trained weights. For egocentric AV action recognition, TBN [48] proposes an architecture to train on audio, RGB frames, and Optical Flow. TIM [8] has modality-specific encoders and a time-interval MLP to query the model at different time-steps. We classify these models as supervised, as they are pre-trained and fine-tuned via linear probing on each dataset. In contrast, contrastively trained self-supervised models like AudioCLIP [35] leverage strong image-text supervision from CLIP [71] to incorporate audio representations into a unified tri-modal embedding space. Similarly, Wav2CLIP [95] applies knowledge distillation from CLIP to audio inputs, effectively mapping audio signals into the same embedding space as text and images. This approach allows for robust audio representations without needing large-scale, audio-specific training from scratch. ImageBind [26] takes a step forward to align six different modalities within a single, rich embedding space. Learning these modalities in tandem supports cross-modal tasks and transfers insights from one modality to another.

**Robustness Benchmarks.** Real-world reliability starts with robustness to distribution shifts [23, 22]. It began with image classification [38, 72, 37], detection [63], segmentation [45, 32], pose estimation [91], and other diverse vision tasks. Existing robustness studies on multimodal models have largely examined video-language retrieval [74], single-source adversaries [96], audio-visual adversarial attacks [86], adaptation methods on vision-language models [11], text-to-image generative models [13], and image-text models [70]. Specifically, [86] investigates the robustness of audio-visual models under adversarial attacks, analyzing how attacks to one or both modalities affect the reliability of fusion strategies. Additionally, MULTIBENCH [55] introduces a large-scale unified benchmark for

multimodal learning. While it includes multiple modalities, including audio and visual, it does not address the robustness of audio-visual models to real-world distributional shifts.

Closest to our motivation, [40] proposes an audio-visual *speech* recognition method with audio and visual corruptions, restricted to speech only. In contrast, AVROBUSTBENCH *specifically targets the robustness of audio-visual recognition models under co-occurring real-world distributional shifts affecting both modalities*. We also note the proposal of two audio-visual benchmarks on VGGSound [10] and Kinetics [47] by READ [97]. Here, unimodal shifts are introduced that are disjoint and unrelated across both modalities.

**Test-Time Adaptation (TTA).** Online TTA focuses on adapting a source/pre-trained model to unlabeled test data. Several TTA methods have been proposed in the literature [90, 44, 84, 9, 77, 14, 67, 99], where adaptation is performed with a single backward pass over each test batch—hence *online*. The seminal work TENT [90] adapts the affine parameters of batch norm layers by minimizing the Shannon entropy [57]. EATA [66] proposes entropy minimization by assigning lower weights to high-entropy samples. While most TTA methods focus solely on single-modality (vision) adaptation, recent works have begun exploring multimodal TTA [81, 17, 61, 69, 36, 46] and segmentation [80]. Notably, READ [97] introduces an audio-visual TTA framework by adapting the QKV parameters of the CAV-MAE joint encoder [30], enabling robust cross-modal fusion under distribution shifts. SuMi [34] performs LayerNorm [4] updates by selecting reliable samples based on multimodal and unimodal entropy, and leverages a mutual information loss between the modalities.

## 3   Proposed Benchmark: AVROBUSTBENCH

We formally introduce AVROBUSTBENCH, a comprehensive suite of 75 AV corruptions (15 corruptions, 5 severities) designed to evaluate AV model robustness under realistic distributional shifts, with a strong focus on real-world deployment, i.e, at test-time. Notably, current audio-visual benchmarks often introduce shifts in one modality [97], treating the shifts *disjoint* from each other. In contrast, AVROBUSTBENCH *simultaneously* applies *co-occurring* corruptions to both modalities, mirroring the *interdependency* that commonly arises in real-world environments.

**Audio-Visual Corruptions.** We introduce 15 diverse audio-visual corruptions at 5 severity levels each, enabling systematic robustness evaluation from mild (1) to extreme (5) [38]. These corruptions fall into three main categories,

- *Digital*: Inspired by ImageNet-C [38] for image corruptions, we adopt the *Gaussian*, *Impulse*, *Shot*, *Speckle*, and *Compression* noises and apply them to video frames. Specifically, we add the JPEG lossy compression to video frames. We borrow this nomenclature from ImageNet-C, with a slight abuse of grouping. Going by the same names for audio, we follow [78] to apply noise by scaling the noise vector based on the signal-to-noise ratio (SNR) and adding it to the audio signal. The SNR controls the severity of corruption, with a lower value indicating more severity. For audio *Compression*, we quantize Discrete Cosine Transform [1] coefficients, adjusting severity through bitrate quantization levels.

- *Environmental*: We introduce *Snow*, *Frost*, *Spatter*, *Wind*, and *Rain* for video frames. We borrow *Snow*, *Frost*, and *Spatter* from ImageNet-C. *Wind* is the same as "Motion Blur" in ImageNet-C. For the effects of *Rain*, we simulate watery and bluish raindrops. In addition, we also introduce *Underwater*, a blue and green tint effect to simulate footage captured by a submerged camera. *Snow* produces a soft, airy sound of falling snowflakes, while *Frost* has a rough, gritty sound. *Spatter* mimics the dripping of water from taps/faucets. *Wind* captures high-speed gusts, and *Rain* reflects hard rainfall. For *Underwater*, we produce a more muffled and submerged noisy audio.

- *Human-Related*: As motivated in Section 1, we also incorporate shifts caused by humans, inspired by various outdoor activities. *Concert* refers to varied brightness effects on the frames and loud music as the noise on the corresponding audio. *Smoke* adds a grayish haze accompanied by the sounds of fire truck sirens and alarms. *Crowd* introduces random human occlusions, such as shadows on frames, while overlaying loud crowd noise, including people talking or cheering. We also introduce *Interference*, where video frames are randomly rotated, and the audio is randomly silenced. This is very indicative of a human fiddling with a recording camera and mic or equipment issues, causing silences at random time intervals.

For audio corruptions in *Environmental* and *Human-Related*, we take recorded environmental samples from Freesound [1] and overlay them onto the audios, adjusting their intensity based on the SNR. *To bring more diversity within the specific corruption, different samples of the dataset have varying patterns, as opposed to a single noise pattern being overlaid in existing images or multimodal benchmarks* [38, 97, 74].

**Datasets.** Our AVROBUSTBENCH comprises four audio-visual benchmark datasets i.e., AUDIOSET-2C, VGGSOUND-2C, KINETICS-2C, and EPICKITCHENS-2C, derived from popular datasets-AudioSet [24], VGGSound [10], Kinetics-Sounds [3], and Epic-Kitchens [16]. *We construct our datasets by introducing our proposed corruptions to the test sets of these datasets,* following the protocols set by [23, 38]. Table 2 provides a summary of the datasets in AVROBUSTBENCH. We further talk about this in the Appendix.

Table 2: Summary and statistics of the datasets comprising AVROBUSTBENCH, after filtering invalid URLs.

| Dataset | # Samples | Classes | Avg. duration |
|---|---|---|---|
| AUDIOSET-2C | 16,742 | 527 | 10 sec |
| VGGSOUND-2C | 14,046 | 309 | 10 sec |
| KINETICS-2C | 3,111 | 32 | 10 sec |
| EPICKITCHENS-2C | 205 | 97 (Noun) 300 (Verb) | 7.4 mins |

## 4 Experimental Settings

**Models.** Our first aim involves studying the robustness at test-time. Our model choices are strictly guided by the availability of publicly accessible codebases and pre-trained weights for AV recognition. On AUDIOSET-2C, VGGSOUND-2C, and KINETICS-2C, we evaluate six SotA models. We utilize three supervised models, i.e., UAVM [28], CAV-MAE [30], and EquiAV [50], explicitly pre-trained and then fine-tuned using source dataset-specific labels. The other three models are contrastively pre-trained on multiple large-scale datasets without task-specific supervision. Wav2CLIP is pre-trained on relatively smaller datasets but can still be adapted effectively for zero-shot AV recognition tasks. ImageBind is primarily a foundational model. Since pre-trained weights for Kinetics-Sounds [3] are not publicly available, we adhere to the recommended training recipes from the cited works. We use the respective trained models for inference on KINETICS-2C. For EPICKITCHENS-2C, we evaluate two SotA supervised models - TBN [48] and TIM [8]. The architectural details and parameters are discussed in the Appendix. As our second aim, we study TTA on our proposed AV robustness benchmark. For a fair comparison, we use pre-trained CAV-MAE as the source model for TTA experiments on VGGSOUND-2C and KINETICS-2C. Table 3 provides a summary.

**Evaluation Metrics.** To iterate, on each dataset, we apply our 15 diverse corruptions at a specific severity level $s$ simultaneously to both the audio and visual modalities. With each corruption defining a task $\mathcal{T}_i$, indexed by $i$, we report a pre-trained model's accuracy on $\mathcal{T}_i$ as $Acc_{i,s}$ = $\frac{\sum_{k=1}^{|\mathcal{T}_i|} \mathbb{1}[y_k=\hat{y}_k]}{|\mathcal{T}_i|}$, where $y_k$ and $\hat{y}_k$ are the $k^{th}$ sample's ground-truth and predicted labels. In addition to task accuracy $Acc_{i,s}$, we evaluate model robustness to distributional shifts using the absolute and relative robustness metrics proposed in [74] and widely adopted in [11, 76, 75]. Given a pre-trained classifier, the accuracy on the source/clean test set is denoted as $A_{cl}$. On the other hand, for $\mathcal{T}_i$ of severity $s$, the accuracy is

Table 3: Details of all the models used in this work.

| Model | # Params. |
|---|---|
| UAVM [29] | 199M |
| CAV-MAE [30] | 191M |
| Equi-AV [50] | 173M |
| TBN [48] | 32.6M |
| TIM [8] | 461M |
| AudioCLIP [35] | 134M |
| ImageBind [26] | 1.2B |
| Wav2CLIP [95] | 314M |

$A_{i,s}$. Defining the drop in accuracy as $\delta A = A_{cl} - A_{i,s}$, the absolute robustness is given by $\alpha_{i,s} = 1 - \frac{\delta A}{100}$. The relative robustness is then given by $\rho_{i,s} = 1 - \frac{\delta A}{A_{cl}}$. $\rho_{i,s}$ and $\alpha_{i,s}$ are bounded between 0 and 1 (both included), with a larger score indicating more model robustness. For AUDIOSET-2C only, for each $\mathcal{T}_i$, we compute the mean average precision (*mAP*) since it is a multilabel dataset. We compute $\rho_{i,s}$ and $\alpha_{i,s}$ accordingly.

**Implementation Details and Remarks.** AVROBUSTBENCH follows the standard evaluation protocol for robustness benchmarks with frozen pre-trained models [38, 23]. For UAVM and CAV-MAE, we use checkpoints trained on AudioSet and VGGSound. Since EquiAV checkpoints were unavailable, we fine-tune its ViT-B/16 [19] encoder on ∼200K VGGSound samples following the original training recipe [2]. Due to computational constraints, we could not reproduce EquiAV on AudioSet (2M training samples). As no pretrained weights exist for Kinetics-Sounds, we train all the supervised

---

[1] https://freesound.org/
[2] https://github.com/JongSuk1/EquiAV

Table 4: *The prevalent challenge is the significant performance gap between each model's clean accuracy and its performance under our proposed audio-visual corruptions at test-time.* We report metrics of models evaluated on AUDIOSET-2C, VGGSOUND-2C, and KINETICS-2C at a severity level of 5. For AUDIOSET-2C, we report the mean of *MAP* across all 15 corruption types (*mMAP*), while for VGGSOUND-2C and KINETICS-2C, we report the mean accuracy (*mAcc*), averaged over corruptions. We also report the drop in performance relative to the clean test set.

| Model | AUDIOSET-2C | | | VGGSOUND-2C | | | KINETICS-2C | | |
|---|---|---|---|---|---|---|---|---|---|
| | mMAP↑ | $\alpha$↑ | $\rho$↑ | mAcc↑ | $\alpha$↑ | $\rho$↑ | mAcc↑ | $\alpha$↑ | $\rho$↑ |
| UAVM [28] | 31.91 (-15.66) | 0.84 | 0.67 | 27.41 (-38.39) | 0.62 | 0.42 | 48.06 (-30.06) | 0.69 | 0.62 |
| CAV-MAE [30] | 31.97 (-17.90) | 0.82 | 0.64 | 35.54 (-29.96) | 0.70 | 0.54 | 58.15 (-29.95) | 0.70 | 0.66 |
| EquiAV [50] | – | – | – | 33.78 (-28.12) | 0.72 | 0.55 | 63.73 (-22.29) | 0.78 | 0.74 |
| AudioCLIP [35] | 12.06 (-16.93) | 0.83 | 0.42 | 11.14 (-15.64) | 0.84 | 0.41 | 23.57 (-27.44) | 0.73 | 0.46 |
| ImageBind [26] | 9.96 (-8.39) | 0.91 | 0.52 | 10.25 (-17.93) | 0.82 | 0.36 | 26.82 (-25.64) | 0.74 | 0.51 |
| Wav2CLIP [95] | 1.74 (-1.40) | 0.98 | 0.55 | 4.99 (-19.33) | 0.81 | 0.21 | 17.25 (-35.40) | 0.65 | 0.33 |

models from scratch. We include AudioCLIP, ImageBind, and Wav2CLIP, where predictions are computed via softmax over the average of audio-text and image-text logits. For models involving text encoders, we use the same prompt template as provided. For EPICKITCHENS-2C, we use pre-trained weights from TBN [48] and TIM [8] for robustness evaluation. For visual inputs, a single video frame is used for CAV-MAE, EquiAV, AudioCLIP, and ImageBind, while UAVM and Wav2CLIP process full videos. Later, we present results for ImageBind demonstrating the invariance of relevant prompt templates during zero-shot inference. For TTA experiments, we follow the recommended settings of the respective works with a batch size of 16. Experiments are performed on NVIDIA A100 and A5000 GPUs. Full model-specific details, hyperparameter settings, and evaluation formulas are provided in the Appendix.

# 5 Results on Robustness at Test-Time

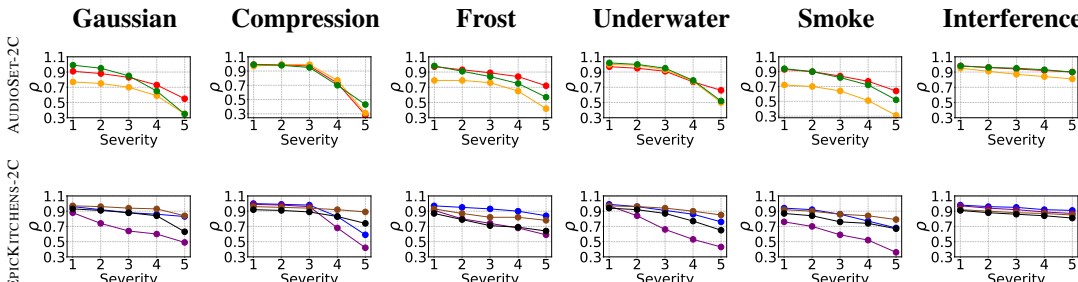

Figure 2: *Corruption severity has a large effect on model robustness; increasing severity decreases robustness.* We illustrate $\rho$ with varying severity on AUDIOSET-2C (top) and EPICKITCHENS-2C (bottom). For AUDIOSET-2C, we show the performance of **CAV-MAE**, **AudioCLIP**, and **ImageBind**. For EPICKITCHENS-2C, we report $\rho$ for **TBN (Noun)**, **TBN (Verb)**, **TIM (Noun)**, and **TIM (Verb)**. The x-axis denotes corruption severity, and the y-axis denotes $\rho$. More examples, including KINETICS-2C, are in the Appendix.

## 5.1 Supervised and self-supervised audio-visual models exhibit low robustness at test-time

Each model is evaluated across our 15 proposed corruption types, or tasks, and we report the mean metrics in Tables 4 and 5. Per-corruption results are provided in the Appendix. We also touch upon subjective evaluations in Appendix.

• **How robust are current AV supervised models at test-time?** On a multilabel dataset like AUDIOSET-2C, models struggle to maintain robustness at a high severity of 5, as reflected in their low absolute $\alpha$ and relative $\rho$ robustness scores. A similar trend is evident in VGGSOUND-2C and KINETICS-2C, with UAVM obtaining the lowest $\rho$. With the shared transformer architecture being queried with one modality at a time, there is no cross-modal information exchange anywhere in the architecture or loss components, leading to poor test-time generalization. CAV-MAE achieves a higher clean performance due to the joint encoder learning rich cross-modal information via concatenated fusion, but the performance breaks down at test-time. The cross-attention between the audio and

visual tokens from the modality-specific transformers is drastically weakened due to severe AV noise, leading to a discrepancy. The joint encoder then fails to infer from such poor associations.

Upon closer inspection on VGGSOUND-2C and KINETICS-2C, EquiAV outperforms CAV-MAE by 2% in $\alpha$ and 1% in $\rho$ and by 8% each, respectively. EquiAV is a "preferred" choice at test-time due to a higher robustness. This is possibly due to learning richer modality-specific representations. The equivariant feature learning setup maps to corresponding transformations in the inter-modal space. This improves the model's ability to maintain a compara-

Table 5: *TBN [48] and TIM [8] also exhibit low robustness to novel AV corruptions* on EPICKITCHENS-2C at a severity level of 5. The drop in performance relative to the clean test set is reported.

tively better robustness, but still fails to generalize well to novel corruptions. At the corruption level (see Appendix), we observe larger performance drops under fine-grained perturbations such as Gaussian, Impulse, etc., which corrupt information at the token level. CAV-MAE's pre-training involves randomly masking 75% of the tokens and contrastively learning on the unmasked ones while reconstructing the masked tokens. During inference, when both modalities are heavily corrupted, a large fraction of tokens becomes unreliable, leading to significant performance degradation.

| Model | EPICKITCHENS-2C | | |
|---|---|---|---|
| | mAcc↑ | $\alpha$↑ | $\rho$↑ |
| TBN [48] (Noun) | 25.68 (-21.66) | 0.78 | 0.54 |
| TBN [48] (Verb) | 52.37 (-13.63) | 0.86 | 0.79 |
| TIM [8] (Noun) | 49.36 (-17.92) | 0.82 | 0.73 |
| TIM [8] (Verb) | 66.55 (-10.55) | 0.89 | 0.86 |

We see that the performance degradation is less pronounced on AUDIOSET-2C. We attribute this to the pre-training data characteristics. AudioSet's 2M unconstrained YouTube clips expose models to natural noise, which makes them more resilient to our proposed corruptions. In contrast, VGGSound and Kinetics-Sounds are smaller, cleaner, and more curated, so models trained on them are less noise-tolerant. Additionally, AUDIOSET-2C uses mAP, a ranking-based metric less sensitive to small score shifts, while VGGSOUND-2C and KINETICS-2C use top-1 accuracy, where even minor rank changes count as full errors, amplifying apparent drops.

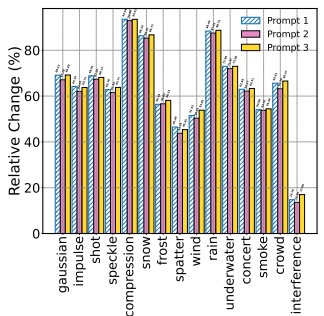

From Table 5, TIM exhibits greater robustness than TBN for both classes. TBN's design is more vulnerable because the fusion within temporal windows may incorrectly associate low-quality features across modalities, leading to error propagation. TIM utilizes a transformer encoder to aggregate long-range cross-modal relations and temporal context, allowing it to potentially "look beyond" corrupted segments. TIM also integrates test-time augmentation by using a sliding window to feed the same interval query with different surrounding contexts to enhance its robustness. On the other hand, both models still suffer performance degradation under severe corruptions. A major reason is that both models use pre-trained feature extractors for individual modalities that may not be robust to multimodal corruptions, providing "corrupted" features as model input before fusion.

Figure 3: *ImageBind's [26] "emergent" zero-shot generalization remains ineffective even with "context"-aware prompts.* We show relative accuracy change (%) i.e., $(\mathcal{A}_{cl} - \mathcal{A}_{i,s})/\mathcal{A}_{cl}$ w/ s=5, on VGGSOUND-2C, using different prompts for the text encoder: **Prompt 1**—a noisy audio of <CLS>.", **Prompt 2**—a noisy photo of <CLS>.", and **Prompt 3**—"a noisy photo of <CLS> and a noisy audio of <CLS>".

• **To what extent do AV self-supervised models maintain robustness under distribution shifts?** ImageBind claims strong zero-shot capability through a unified embedding space; we observe otherwise. The ViT-based image encoder and the Transformer-based audio encoder, both trained using the InfoNCE loss [68], face significant challenges in effectively associating corrupted (image, audio) pairs. This can be seen from the large gap in *Clean* and *mAcc* and low values of $\alpha$ and $\rho$, indicating poor robustness. InfoNCE encourages instance discrimination within a shared embedding space rather than focusing on capturing robust cross-modal associations. This makes the learned representations highly sensitive to noise. It shows limited robustness despite being pre-trained on multiple distributions, including AudioSet and VGGSound. Similarly, AudioCLIP and Wav2CLIP contrastively learn associations by distilling knowledge from CLIP [71]. However, from Table 4, we observe that such models struggle to infer based on new cross-modal associations arising from corruptions. To specify, Wav2CLIP has been pre-trained on VGGSound but subsequently fails at test-time, with a mean accuracy drop of 15.64%. Similarly, AudioCLIP has been pre-trained on AudioSet but struggles at test-time, with a 16.93% gap. Another contributing factor is the low

robustness of CLIP to visual corruptions, as shown in [61]. However, if we were to choose a model solely based on *mAcc* and $\rho$, ImageBind would be relatively better.

- **Robustness at test-time decreases with an increase in corruption severity.** As one would expect, model robustness begins to decline with an increase in AV corruption severity (Figure 2). In the Appendix, we illustrate more examples. Notably, all models exhibit a steady decline in robustness as the severity increases. However, for *Interference*, models show a steady drop to almost being stable in robustness with increasing severity. This corruption appears easier for models, as recognition remains feasible even when video frames are heavily rotated and audio is silenced. This is also reflected in *Acc*. On *Compression* in AUDIOSET-2C, AudioCLIP and CAV-MAE show the lowest $\rho$ at severity level 5, while TBN's performance drops significantly beyond severity 3. Despite frames being visually clear and audio audible to humans, these models struggle to recognize them. Similar trends are observed for TBN (Verb) and TBN (Noun). Zero-shot models exhibit notably lower robustness. On average, models are less robust to *Digital* corruptions and more robust to *Human-Related* ones, except AudioCLIP.

- **Prompt tweaks yield negligible gains for ImageBind.** The findings of ImageBind suggest that the shared image-centric embedding space enables effective cross-modal recognition, even when the image is not used as the anchor during inference. The natural question is, would effective prompt design enhance audio-text representations (or image-text), thereby enhancing the overall audio-visual performance? We analyze the effect of prompts, as defined in Figure 3, that would provide a "stronger" contextual grounding for audio-text, image-text, or joint audio-visual alignment. Except for *Interference*, across corruption types, prompt variations yield minimal performance gains, while the high relative accuracy changes highlight the limited robustness of ImageBind's text encoder. The reason is simple - the text features are unaware of or independent of the noisy audio and image features, leading to low similarity between the embeddings. To drive further discussion, prompt selection at test-time, is unsuitable, impractical, unknown, and time-consuming [61].

## 6 Online TTA Results

**Motivation.** As discussed earlier, online TTA focuses on adapting pre-trained models to sequential, unlabeled test batches under distributional shifts [90] with a single iteration on each due to privacy and memory constraints [62]. We evaluate READ [97] and SuMi [34], two SotA approaches for online AV TTA. We also adapt other online TTA approaches [90, 73, 66, 67] following READ and SuMi, and report the results in Table 6. Our proposed AV2C adapts the QKV attention weights and minimizes a weighted Shannon entropy, with a larger weight on low-entropy samples [66].

- **Existing TTA methods generally struggle to adapt to the bimodal AV corruptions.** TENT [90], RPL [73], and SAR [67] update all the norm parameters based on proposed loss functions and show comparable to reduced performances compared to the CAV-MAE source model. Norm updates have been effective under domain shifts for a single modality [54]. However, for severe AV corruptions, it leads to overfitting of the norm parameters (LayerNorm) on the test data, causing unstable adaptation. Specifically, AV corruptions under *Digital* (Gaussian, Impulse, etc) result in larger performance drops. These introduce pixel-level noise in video frames and also significantly distort the fundamental frequency in audio signals, thereby disrupting essential low-level features, leading to noisier predictions. EATA [66] possibly benefits from adaptation solely based on the larger emphasis on low-entropy samples in a test batch.

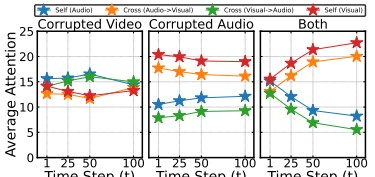

Figure 4: *Over time steps (t) during online TTA, an attention imbalance in the form of modality bias begins with AV corruptions, leading to a degrading performance of READ.* Average attention weights are computed across 12 heads from 1 block of CAV-MAE's joint encoder for a batch size of 64. The numbers indicate averaged attention, scaled by 10,000. We show *Gaussian* on VGGSOUND-2C for discussion.

In prior methods, the affine parameters of LayerNorm [4] in all the attention blocks are updated along the feature dimension, and have minimal direct relation with cross-modal fusion during TTA. SuMi [34] first applies interquartile range smoothing to identify samples within the 25%–75% confidence range, and then selects low-entropy samples using multiple thresholds based on unimodal and multimodal entropy. To balance information exchange between modalities during adaptation, a mutual information loss is employed to update the LayerNorm parameters. However, SuMi achieves

Table 6: *There still exists a wide gap between mean accuracy by TTA baselines and the source model's accuracy on VGGSound (65.50%) and Kinetics-Sounds (88.10%).* CAV-MAE [30] is the source model initialized by VGGSound/Kinetics-Sounds weights'. We report mean accuracy (%) on VGGSOUND-2C (top) and KINETICS-2C (bottom). We evaluate TTA methods at severity 5 with a batch size of 16 for VGGSOUND-2C and KINETICS-2C. Source denotes the direct inference of CAV-MAE.

| | TTA Method | Gaussian | Impulse | Shot | Speckle | Compression | Snow | Frost | Spatter | Wind | Rain | Underwater | Concert | Smoke | Crowd | Interference | Mean |
|---|---|---|---|---|---|---|---|---|---|---|---|---|---|---|---|---|---|
| **VGGSOUND-2C** | Source [30] | 20.39 | 23.73 | 20.72 | 25.34 | 17.26 | 25.07 | 46.82 | 48.46 | 50.17 | 29.89 | 42.19 | 47.61 | 32.93 | 47.71 | 54.88 | 35.54 |
| | TENT [90] | 1.04 | 1.51 | 1.15 | 2.81 | 2.51 | 1.84 | 7.34 | 47.53 | 49.13 | 2.71 | 29.55 | 40.03 | 10.53 | 35.45 | 53.16 | 19.09 |
| | SAR [67] | 1.89 | 3.30 | 1.96 | 7.65 | 5.57 | 3.74 | 49.81 | 50.67 | 51.99 | 6.04 | 46.65 | 42.64 | 18.28 | 45.35 | 55.55 | 26.07 |
| | RPL [73] | 1.13 | 1.54 | 1.18 | 3.01 | 2.66 | 2.11 | 14.70 | 49.69 | 50.62 | 2.94 | 46.41 | 48.28 | 11.08 | 44.61 | 54.28 | 22.28 |
| | EATA [66] | 37.20 | 36.53 | 36.71 | 34.89 | 25.60 | 38.42 | 49.28 | 50.80 | 51.76 | 42.38 | 46.87 | 50.39 | 37.05 | 52.36 | 54.54 | 42.98 |
| | READ [97] | 38.30 | 26.11 | 37.60 | 19.98 | 12.88 | 26.71 | 49.47 | 51.51 | 52.93 | 35.14 | 25.67 | 50.83 | 47.09 | 52.85 | 53.99 | 38.74 |
| | SuMi [34] | 22.24 | 23.54 | 22.07 | 25.52 | 17.11 | 24.23 | 46.58 | 48.33 | 50.01 | 29.40 | 41.74 | 47.51 | 32.71 | 47.51 | 54.86 | 35.56 |
| | AV2C | 38.34 | 37.32 | 37.65 | 32.47 | 21.18 | 40.78 | 50.13 | 52.33 | 53.60 | 43.98 | 46.51 | 51.10 | 46.74 | 53.90 | 54.84 | 44.06 |
| **KINETICS-2C** | Source [30] | 51.34 | 48.82 | 51.27 | 46.90 | 44.88 | 47.88 | 59.97 | 63.16 | 68.76 | 58.54 | 61.51 | 66.80 | 48.15 | 74.81 | 79.44 | 58.15 |
| | TENT [90] | 42.45 | 40.95 | 43.84 | 51.72 | 26.61 | 48.45 | 67.83 | 63.89 | 74.45 | 66.65 | 59.61 | 71.69 | 56.11 | 78.93 | 81.29 | 58.30 |
| | SAR [67] | 25.94 | 25.34 | 28.17 | 33.40 | 15.55 | 43.18 | 59.79 | 46.10 | 63.46 | 51.54 | 43.52 | 53.79 | 46.33 | 53.73 | 63.71 | 43.57 |
| | RPL [73] | 45.03 | 43.79 | 46.18 | 52.14 | 28.21 | 50.12 | 67.47 | 64.52 | 74.61 | 66.45 | 60.09 | 72.12 | 56.12 | 79.05 | 81.61 | 59.17 |
| | EATA [66] | 50.39 | 49.81 | 49.99 | 51.96 | 43.04 | 55.02 | 65.34 | 66.41 | 73.17 | 64.61 | 61.48 | 71.79 | 53.85 | 78.94 | 81.60 | 61.16 |
| | READ [97] | 54.00 | 52.43 | 54.43 | 51.90 | 50.86 | 55.83 | 64.99 | 69.50 | 71.34 | 64.03 | 63.57 | 69.07 | 55.28 | 77.32 | 79.96 | 62.30 |
| | SuMi [34] | 49.41 | 48.89 | 49.34 | 51.79 | 42.29 | 55.06 | 65.02 | 66.24 | 73.06 | 65.07 | 61.14 | 71.74 | 54.05 | 78.92 | 81.49 | 60.94 |
| | AV2C | 52.37 | 51.28 | 51.91 | 52.02 | 46.7 | 56.53 | 67.09 | 68.38 | 73.35 | 65.65 | 60.75 | 72.54 | 55.64 | 79.31 | 81.38 | 62.33 |

only negligible improvements, as it does not explicitly encourage interactions between noisy modality tokens. In addition, the need for multiple thresholds and extensive hyperparameter tuning significantly hampers its practicality for real-time TTA. In contrast, READ [97] adapts the QKV attention weights in the joint encoder, enabling dynamic self- and cross-attention between audio and visual tokens. It performs well under unimodal corruptions by leveraging the cleaner modality for reliable fusion, as seen in Tables 7 and 8. Notably, in VGGSound, audio contains more task-relevant information [10, 97] and vice-versa on Kinetics-Sounds [3]. With our proposed AV corruptions, READ's effectiveness declines as both noisy audio and visual tokens hamper the self- and cross-attention dynamics (Figure 4), impairing reliable cross-modal fusion. We see, with corrupted audio only, the cross-attention from visual to audio begins to increase with time-step $t$. However, when both modalities are corrupted, there is an increasing *modality bias* towards the visual tokens from audio (13.09 at $t$=0 to 20.04 at $t$=100) that compounds with time and results in sub-optimal performances on each corruption. Batches in the future are affected by this modality bias. This growing attention imbalance likely contributes to READ's performance drop.

• **Discussions on our proposed TTA method AV2C.** With encouraging directions from EATA, our proposed AV2C benefits on VGGSOUND-2C and obtains comparable performance on KINETICS-2C. On VGGSOUND-2C with 309 classes, where model uncertainty can be higher, penalizing high-entropy samples that hurt adaptation and selecting diverse samples for entropy minimization to instead update the QKV attention weights can be fruitful. Selective modality tokens, as the input to CAV-MAE's joint encoder, contribute more to updating the attention weights for efficient cross-modal fusion of the current test batch, leading to improvements of 5.32% and 8.5% over READ and SuMi, respectively. *The mathematical details of the proposed AV2C can be found in the Appendix.*

Table 7: Unimodal corruption analysis on VGGSOUND-2C. $\Delta_V$, $\Delta_A$, and $\Delta_{AV}$ denote the drop from clean accuracy accordingly.

| TTA Method | Corrupted Video | Corrupted Audio | Both | $\Delta_V, \Delta_A, \Delta_{AV}$ |
|---|---|---|---|---|
| Source | 56.04 | 50.48 | 35.54 | -9.46, -15.02, -29.94 |
| TENT | 55.62 | 28.90 | 19.09 | -9.88, -36.60, -46.41 |
| READ | 55.18 | 53.29 | 38.74 | -10.32, -12.21, -26.76 |
| SuMi | 55.99 | 50.18 | 35.56 | -9.51, -15.32, -29.94 |
| AV2C | 55.23 | 54.76 | 44.06 | -10.27, -10.74, -21.44 |

Table 8: Unimodal corruption analysis on KINETICS-2C. $\Delta_V$, $\Delta_A$, and $\Delta_{AV}$ denote the drop from clean accuracy accordingly.

| TTA Method | Corrupted Video | Corrupted Audio | Both | $\Delta_V, \Delta_A, \Delta_{AV}$ |
|---|---|---|---|---|
| Source | 75.46 | 81.03 | 58.15 | -12.64, -7.07, -29.95 |
| TENT | 76.41 | 81.52 | 58.30 | -11.69, -6.58, -29.80 |
| READ | 77.02 | 82.31 | 62.30 | -11.08, -5.79, -25.80 |
| SuMi | 76.17 | 81.32 | 60.94 | -11.93, -6.78, -27.16 |
| AV2C | 77.21 | 81.21 | 62.33 | -10.89, -6.89, -25.77 |

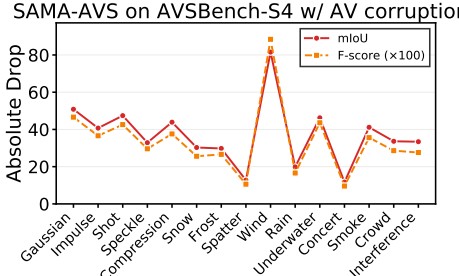

Figure 5: *State-of-the-art audio-visual segmentation models still struggle in the presence of bimodal audio and visual corruptions.* We use SAMA-AVS [58] to directly infer on the AVSBench-S4 [101] test set with our proposed corruptions (severity 5). Each task includes 740 videos, and we report the absolute drops in mean intersection over union (mIoU) and F-score relative to the clean AVSBench-S4 results of SAMA-AVS.

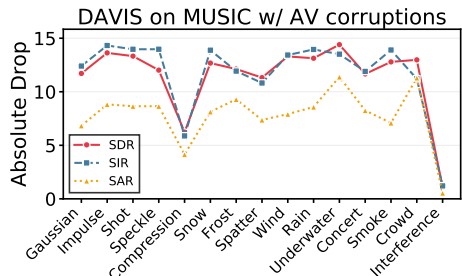

Figure 6: *Visually-guided sound source separation becomes challenging under audio and visual corruptions.* We employ a pre-trained DAVIS [41] on the MUSIC test set [100] (250 videos) with our AV corruptions (severity 5). The y-axis shows the absolute drops in Signal to Distortion Ratio (SDR), Signal to Interference Ratio (SIR), and Signal to Artifact Ratio (SAR) compared to DAVIS' performance on the clean MUSIC test set.

## 7 Experiments on other Audio-Visual Downstream Tasks

In this section, beyond recognition, we present results and discuss robustness on other downstream tasks like audio-visual segmentation (AVS) [101, 58, 53] and sound source separation [20]. In the Appendix, we present audio-visual retrieval results where we show that audio-visual correspondences are drastically hampered, resulting in low recall scores.

● **Audio-Visual Segmentation.** We employ the SotA pre-trained SAMA-AVS [58] and introduce our proposed corruptions, at a severity of 5, to both the modalities of the AVSBench-S4 [101] test set, with 740 videos in each task. In Figure 5, we present the absolute drops in mean Intersection over Union (mIoU) and F-score relative to the clean AVSBench-S4 performance achieved by SAMA-AVS. For context, the mIoU and F-score of SAMA-AVS on clean AVSBench-S4 are 81.553 and 0.886, respectively. As seen, SoTA AVS models like SAMA-AVS still struggle in the presence of bimodal corruptions.

● **Sound Source Separation.** For this crucial task, we use the SotA DAVIS [41] on the MUSIC test set [100] with our proposed audio-visual corruptions. Each task has 250 videos. In Figure 6, we illustrate the absolute drops in Signal to Distortion Ratio (SDR), Signal to Interference Ratio (SIR), and Signal to Artifact Ratio (SAR) relative to DAVIS's performance on the clean MUSIC test set (higher is better). For comparison, the SDR/SIR/SAR on the clean MUSIC test set are 11.68/18.36/15.26, and the mean noisy scores are 0.18/6.77/7.46, respectively. To conclude, SoTA sound source separation models struggle in the presence of bimodal corruptions.

## 8 Conclusion

We introduce AVROBUSTBENCH, a benchmark for evaluating the test-time robustness of audio-visual recognition models. AVROBUSTBENCH comprises four audio-visual datasets, AUDIOSET-2C, VGGSOUND-2C, KINETICS-2C, and EPICKITCHENS-2C, each augmented with bimodal audio-visual corruptions that are both *co-occurring* and *correlated* across modalities. We conduct a comprehensive analysis of model robustness under these challenging distribution shifts. Furthermore, we evaluate a suite of online TTA methods, offering key insights and revealing their critical limitations. We also propose a simple yet effective TTA baseline that outperforms existing methods on the benchmark. We hope that AVROBUSTBENCH will facilitate deeper understanding and drive future research on robust, adaptable audio-visual systems in real-world settings.

## Acknowledgments

This project was supported by a grant from UT Dallas and an NVIDIA Academic Grant Program Award. This research is also partially supported by the National Science Foundation (NSF) under Grant No. 2513070. We also thank Jia Li and Weiguo Pian for helpful discussions.

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

# A Appendix

In this document, we provide additional insights, experimental results, and hold other discussions on AVROBUSTBENCH. We organize all of this as follows,

1. Section A.1 describes, in great detail, the benchmark datasets we propose i.e., AUDIOSET-2C, VGGSOUND-2C, KINETICS-2C, and EPICKITCHENS-2C. We also describe the implementation details of our proposed real-world audio-visual corruptions, and show visuals of EPICKITCHENS-2C.

2. In Section A.2, we dive deep into the architectures and implementation details of all the supervised and self-supervised audio-visual models that are used for our study. We discuss the training settings of supervised models on Kinetics-Sounds, which is then used for evaluation purposes on KINETICS-2C. We also give details of the online TTA methods that are used.

3. We give a complete formulation of AV2C in Section A.3. Section A.4 has other detailed results from the main paper. We also touch upon other experiments-a subjective test on humans (Section A.5), audio-visual retrieval (Section A.6), and the recognition performance of audio-visual large language models (Section A.7).

## A.1 Proposed Benchmark: AVROBUSTBENCH

### A.1.1 Datasets

As mentioned in the main paper, AVROBUSTBENCH consists of four benchmark audio-visual datasets, AUDIOSET-2C, VGGSOUND-2C, KINETICS-2C, and EPICKITCHENS-2C, constructed from the test sets of popular audio-visual datasets: AudioSet [24], VGGSound [10], Kinetics-Sounds [3], and Epic-Kitchens [16], respectively. These datasets span diverse domains, environments, and action categories, offering a broad and realistic evaluation suite for audio-visual recognition models. AudioSet is one of the largest audio-visual datasets in terms of training samples. It is released as YouTube URLs, and after filtering out invalid URLs, AUDIOSET-2C contains 16,742 audio-video test pairs. Each clip is roughly 10s and spans 527 classes. Due to its multilabel nature, mean average precision (mAP) is the standard evaluation metric. VGGSound consists of roughly 10s of YouTube videos spanning 309 classes, including human actions. After filtering invalid URLs, VGGSOUND-2C contains 14,046 test pairs. From Kinetics-Sounds' test set, we construct KINETICS-2C, comprising YouTube videos capturing a diverse range of human actions. KINETICS-2C contains 3,111 clips across 32 classes, each around 10s long. EPICKITCHENS-2C is the corrupted test set of Epic-Kitchens that has 205 egocentric video clips capturing daily kitchen tasks of an average duration of 7.4 mins each. We follow the protocol, as set by [16], for action evaluation. Each action is uniquely defined by a combination of a "Verb" and a "Noun". In the main paper, we give a summary of the number of samples and classes.

We adopt the standard evaluation protocol for robustness, as outlined in [23, 38], and introduce real-world audio-visual corruptions that are applied *simultaneously* to both modalities during testing. Each corruption type is used with a specific severity level sampled from a fixed scale (typically 1–5), ensuring consistency across evaluations. Visual corruptions are applied to every frame in the video, while audio corruptions are added directly to the video's corresponding audio waveform.

The corruptions are chosen to reflect real-world challenges. They are designed to be *co-occurring* and *correlated*, mimicking the natural interplay of noise that might affect both modalities in deployment settings like autonomous vehicles or wearable devices. To facilitate further research, we also release the code, enabling easy reproducibility and extension of the benchmark.

### A.1.2 Implementation of Audio-Visual Corruptions

Here, we provide the implementation details of the audio-visual corruptions. As mentioned earlier, we group the 15 corruptions, each spanning 5 severity levels, into three categories, i.e., *Digital*, *Environmental*, and *Human-Related*. In total, we propose 75 audio-visual corruptions.

- *Digital:* For visual corruptions, we adopt the exact implementations of *Gaussian*, *Impulse*, *Shot*, and *Speckle* from ImageNet-C [38]. For *Compression*, we utilize the JPEG-based

compression proposed in the same work. Throughout, we apply audio corruptions at signal-to-noise ratios (SNRs) ranging from 40 to 0 in intervals of 10, where a lower SNR corresponds to higher corruption severity. For example, a severity level of 5 indicates an SNR of 0. In the case of *Gaussian*, we generate a noise vector matching the shape of the audio signal, sample it from a standard normal distribution, scale it according to the desired SNR, and add it to the original audio waveform. For *Impulse*, we sample a salt-and-pepper noise vector based on a uniform random mask. For *Shot*, zero-mean Poisson noise is derived from the normalized audio waveform. In *Speckle*, we multiply zero-mean Gaussian noise element-wise with the audio waveform to create speckle distortions. For each corruption type, the noise is scaled using the audio signal power $P_{sig}$ and the raw noise power $P_n$, both computed as the mean squared amplitude of their respective signals. The noise scaling factor $\beta$ is computed as $\sqrt{\frac{P_{sig}}{10^{SNR/10}*P_n}}$, making sure that the noise meets the desired SNR, i.e., severity. Then, the scaled noise ($\beta$·noise) is added back to the original audio waveform. For audio *Compression*, we control the severity based on the bitrate quantization levels, computed as $2^c$ where $c \in [24, 16, 8, 4, 2]$. A severity of 5 would refer to bitrate levels of 4. We split the mono waveform into fixed-size blocks of size 1024, apply an orthonormal DCT to each block, normalize, and quantize its coefficients to the required bitrate level. We then reconstruct the audio waveform via inverse DCT before concatenating the blocks.

- *Environmental:* The visual corruptions in *Snow*, *Frost*, and *Spatter* are directly taken from the implementation in [38]. For *Wind*, we use the implementation of "Motion blur", as in the same work. For the visual effects of *Rain* on video frames, we control the severity based on droplet density, scale, zoom factor, threshold, motion-blur settings, and blend weight. A monochrome rain mask is generated by sampling Gaussian noise, applying a clipped zoom to cluster droplets, and thresholding to isolate individual raindrops. We also add a tinted bluish color by expanding it into the RGB channels with custom scaling factors. Similarly, for *Underwater*, we control the severity based on Gaussian blur kernel size, red-channel attenuation, contrast reduction, and haze intensity. These are used to mimic light absorption and scattering underwater. We first reduce the red channel by the red-channel attenuation factor to simulate the color shift, then apply a Gaussian blur to soften edges as light diffuses. Additionally, the contrast is lowered via linear scaling, and a semi-transparent white haze overlay, based on the haze intensity, is blended in.

  As mentioned in the main text, we borrow recorded samples from Freesound for the audio corruptions. We ensure that their sampling rates match those of the target audio and are converted to mono. Each corruption is overlaid directly onto the waveform, with its intensity precisely controlled by the specified SNR (severity), as in the case of Digital. To introduce diversity within each corruption type, we avoid using a fixed noise pattern across all audio samples. Instead, for every corruption, we randomly select one noise sample from a pool of $N$ options, where $N \in [15, 5, 8, 8, 8, 31]$ for *Snow*, *Frost*, *Spatter*, *Wind*, *Rain*, and *Underwater*, respectively.

- *Human-Related:* We introduce human-level corruptions that closely reflect real-world conditions. In the *Concert* setting, we adopt the "Brightness" visual effect from ImageNet-C, and overlay loud music samples from Freesound as the audio corruption, with severity controlled via the SNR. For *Smoke*, we simulate grayish visual effects by generating a Gaussian-blurred noise map, scaled by a factor to control the standard deviation and replicated across RGB channels. Corresponding audio corruptions use recorded smoke alarms and loud sirens from FreeSound. For *Crowd*, we project random human occlusions onto a video frame. The size of the occlusion denotes the severity of it. We place it at a random location, and blend it over the original frame. For an audio effect, we overlay random crowd noises from FreeSound. For *Interference*, we randomly rotate a video frame with the angle sampled from $(-(6 \times \alpha + 5), +(6 \times \alpha + 5))$ in degrees where $\alpha$ is the severity. So, for an $\alpha$ of 5, a video frame could be randomly rotated between -35 to 35 degrees. We randomly silence a fraction of the audio (a higher severity denotes more silencing) between $[0.1, 0.2, 0.3, 0.4, 0.5]$. Throughout this group, we make sure that, for each introduced audio corruption, we have diverse noise patterns within the same task/corruption as in *Environmental*.

The full code implementation is released here: https://github.com/sarthaxxxxx/AV-C-Robustness-Benchmark/tree/master

### A.1.3 Visualizations

In Figure 7, we showcase sample visual corruptions from EPICKITCHENS-2C. For the full audio-visual experience, we urge the reader to see and hear the difference in action by watching our demo on YouTube: https://www.youtube.com/watch?v=hYdcRO3BuIY.

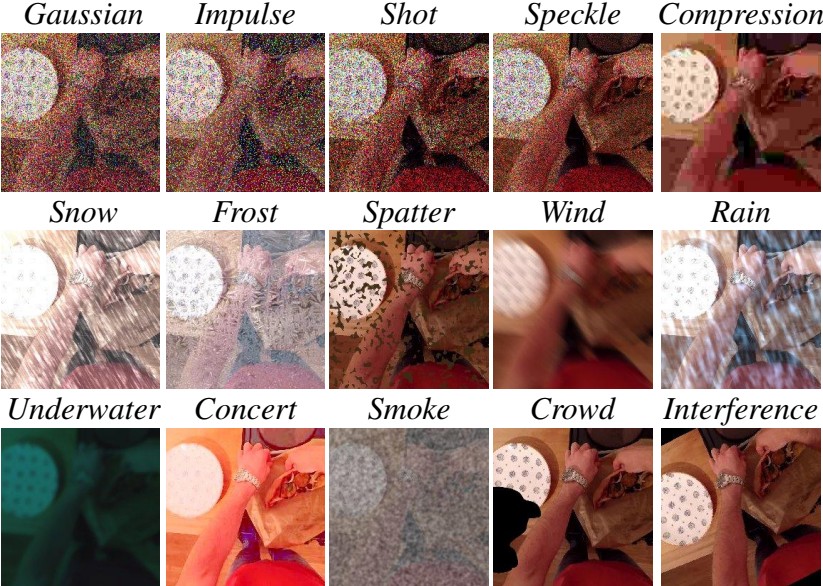

Figure 7: We sample a random video frame from Epic-Kitchens [16] and show visualizations of the proposed 15 audio-visual corruptions on it, at a severity level of 5. This constitutes EPICKITCHENS-2C.

## A.2 Implementation Details

### A.2.1 Models

On AUDIOSET-2C and VGGSOUND-2C, we directly infer using *supervised* models like UAVM [28], CAV-MAE [30], and EquiAV [50]. UAVM employs modality-specific transformers that process audio and video features in parallel. Audio and video features are extracted via ConvNeXt-Base [59] backbones for spectrograms and frames, respectively, before being fed into their respective transformers. A shared transformer is then applied twice (once per modality), and the resulting logits are averaged, followed by a softmax to produce final predictions. CAV-MAE has separate audio and visual encoders that process spectrograms and a randomly sampled video frame, respectively, followed by a joint encoder trained through contrastive learning on audio-video masked tokens. Since pre-trained weights for Kinetics-Sounds [3] are not publicly available, we adhere to the recommended training recipes from the cited works. We then use the respective trained models for inference on KINETICS-2C. The training details are provided later. In addition to these models, we also *self-supervised* models like AudioCLIP [35], ImageBind [26], and Wav2CLIP [95]. Since our focus would be on their respective audio and visual encoders, this would allow us to gauge their robustness to the proposed corruptions. For experiments on EPICKITCHENS-2C, we conduct evaluations using TIM [8] and TBN [48], as supervised models, following their official methodologies. TIM processes both modalities through separate encoder streams before applying cross-modal attention to capture the intricate relationships between sounds and visual actions typical in kitchen environments. We use the same feature extraction pipeline with Omnivore [27] and VideoMAE-L [87] for visual features and Auditory SlowFast [49] for audio features. The model encodes time intervals from each modality as queries to identify actions occurring during specific timeframes. TBN performs mid-level fusion of video frames and audio within temporal binding windows. We corrupt the frames and audio during inference time and feed them directly into the model for prediction.

**Predictions from self-supervised models.** For AudioCLIP and ImageBind, we sample a single frame from the video to extract $f_v$, while Wav2CLIP operates over a tensor of video frames. Audio features $f_a$ are extracted accordingly, ensuring sampling rates align with each model's specifications. To obtain the text features $\{f_{t,c}\}_{c=1}^C$, where $f_{t,c}$ is the text feature of class $c$ out of $C$ classes, we compute the audio-text ($S^{a,t}$) and image-text logits ($S^{v,t}$) as,

$$S^{a,t} = \frac{<f_a, f_{t,c}>}{||f_a||_2 \cdot ||f_{t,c}||_2} \quad , \quad S^{v,t} = \frac{<f_v, f_{t,c}>}{||f_v||_2 \cdot ||f_{t,c}||_2} \tag{1}$$

We then compute $\frac{S^{a,t}+S^{v,t}}{2}$ to obtain the averaged logits of class $c$. Applying a softmax operation over all classes yields the final likelihood of each audio-visual pair.

**Prompt Templates.** We use the default prompt templates provided with each zero-shot model. In the main paper, we also show that different prompt templates for ImageBind yield minimal improvements.

### A.2.2 Training settings of supervised models for KINETICS-2C

**CAV-MAE** consists of 11 transformer layers per modality to extract features from both audio and visual inputs. 10 frames are sampled from each clip, and one frame is randomly selected as input to the visual transformer encoder. For the audio stream, the 10 s waveform is converted into a spectrogram, which is then passed through the audio transformer encoder. During the fine-tuning phase on Kinetics-Sounds, following the CAV-MAE setup [3], we freeze the pretrained visual and audio encoders from the pretrained model CAV-MAE-Scale++ and add a randomly initialized MLP classifier on top with 32 classes. The resulting fine-tuned model is treated as the source model for experiments in online TTA. For input normalization, we set the dataset mean to -5.081 and the standard deviation to 4.4849, following [97]. We use a learning rate of 1e-4, a batch size of 48, and train for 10 epochs.

During pretraining, **EquiAV** processes 10 s video clips, where the visual stream involves sampling frames and applying spatial augmentations, while the audio stream is converted into spectrograms and undergoes temporal augmentations. We use the AudioSet-2M pretrained model as the backbone and add newly initialized layer for the fine-tuning stage. We fine-tune it on the Kinetics-Sounds training set with both audio and visual data under multi-modal mode. The model is trained for a maximum of 50 epochs with a learning rate of 1e-4. For input normalization, we use the same dataset mean and standard deviation as in CAV-MAE.

In **UAVM**, audio features are extracted using an AudioSet-2M pretrained ConvNeXt-Base, while visual features are obtained using the official ImageNet-pretrained ConvNeXt-Base. UAVM model consists of three modality-specific Transformer layers, followed by three shared Transformer layers. For each input sample, a separate forward pass is performed for the audio and visual modalities, and the predictions from the two passes are averaged to produce the final fused prediction. We use a learning rate of 1e-4, a batch size of 144, and train for 10 epochs. During training, each iteration uses only one modality, with a 50% chance of selecting either audio or video.

### A.2.3 TTA settings

**Source [30]** Following the audio-visual TTA protocol set by [97], we use CAV-MAE [30] as the source model. For experiments on VGGSOUND-2C and KINETICS-2C, as the initialization, we use pre-trained weights from VGGSound and Kinetics-Sounds (as mentioned in A.2.2), respectively, and do a direct inference on a test-batch.

**TENT [90]** Following [90, 97], all the LayerNorm parameters of the CAV-MAE audio, visual, and joint encoder are updated by minimizing the Shannon entropy [79] of model predictions. For VGGSOUND-2C and KINETICS-2C, we optimize with Adam using a learning rate of 1e-4.

**RPL [73]** The LayerNorm parameters of the CAV-MAE model are updated using the generalized cross-entropy loss. We use an Adam optimizer and a learning rate of 1e-4.

**EATA [66]** On VGGSOUND-2C and KINETICS-2C, we use an Adam optimizer with a learning rate of 1e-4. The entropy threshold is set $0.4\times \log(C)$, where C refers to the number of class labels.

---

[3]https://github.com/yuangongnd/cav-mae

Since the source data is unavailable, we do not use the Fisher regularization to minimize forgetting of source domain knowledge.

**SAR [67]** The LayerNorm parameters of CAV-MAE are updated using the Adam optimizer with a fixed learning rate of 1e-4. For stable entropy minimization, we adopt the same confidence threshold as in EATA [66], while a threshold of 0.2 is used for model recovery. The exponential moving average (EMA) coefficient for model predictions is set to 0.9.

**READ [97]** Following their original implementation, the QKV parameters of CAV-MAE's joint encoder are self-adapted based on a confidence-aware and balancing loss function. A confidence threshold of $\frac{1}{e}$ is used. We use the Adam optimizer with a learning rate of 1e-4 for both VGGSOUND-2C and KINETICS-2C.

**SuMi [34]** LayerNorm parameters are updated based on Adam using learning rates of 1e-4 and 1e-5 for KINETICS-2C and VGGSOUND-2C respectively. The multimodal threshold [66] is set to $0.4\times$ log(C). As per their recommendation, a mutual information loss is applied for every half iteration. All other dataset-specific hyperparameters are set based on the original work.

Note: To be uniform, all of our TTA experiments are done with a batch size of 16.

### A.3 AV2C - **Our proposed TTA method**

Our proposed online TTA framework, AV2C, consists of two key areas. The first area emphasizes efficient audio-visual cross-modal fusion at test-time. Let $f_a$ and $f_v$ represent the audio and visual embeddings, respectively, obtained from the modality-specific encoders of the CAV-MAE source model [30]. To enable fine-grained integration across modalities at the token level, we concatenate these embeddings to form a joint representation: $f_{av} = [f_a; f_v]$

Inspired by READ [97], a simple proposal for good, reliable, and on-the-fly fusion is to modulate the attention parameters of the joint-encoder, which dynamically re-weights modality contributions, enabling more robust integration under distribution shifts. Formally speaking, let $w_q$, $w_k$, and $w_v$ denote the weight matrices of the query, key, and value parameters of the attention block in the joint encoder. And, let $b_q$, $b_k$, and $b_v$ be their corresponding biases. So, the attention matrices are,

$$\mathcal{Q} = f_{av}W_q + b_q \tag{2}$$
$$\mathcal{K} = f_{av}W_k + b_k \tag{3}$$
$$\mathcal{V} = f_{av}W_v + b_v \tag{4}$$

Throughout, $\mathcal{Q}$, $\mathcal{K}$, and $\mathcal{V}$ are adapted at test-time with all other model parameters being frozen and fixed to the default source weights. With $\mathcal{Q}$, $\mathcal{K}$, and $\mathcal{V}$ being adaptive, both self- and cross-attention are computed at the token level to dynamically capture and integrate modality-specific and modality-shared information, enabling robust fusion under distribution shifts.

However, under simultaneous distributional shifts in both modalities, the input token quality may degrade significantly, resulting in unreliable attention computations and elevated model uncertainty. To address this, in the second area, we adapt the attention parameters $\mathcal{Q}$, $\mathcal{K}$, and $\mathcal{V}$ at test time by selectively updating them based on confident predictions. Inspired by the loss formulation in [66], we apply entropy minimization but only on low-entropy (i.e., high-confidence) samples. This selective update strategy ensures stable and reliable adaptation without propagating noise from uncertain predictions.

We minimize a weighted Shannon entropy [79] of model predictions, as an unsupervised objective. That is, the optimization based on the entropy $H(x)$ at time-step $t$ is,

$$\underset{\theta}{\operatorname{argmin}} \ - \widehat{\eta(x)} H(x) \tag{5}$$

$$= \underset{\theta}{\operatorname{argmin}} \ - \widehat{\eta(x)} \sum_{c\in\mathcal{C}} p(y_t = c \mid x) \log p(y_t = c \mid x) \tag{6}$$

where, $C$ is the complete set of classes and $p(y_t|x)$ is the probability of output logits. $\widehat{\eta(x)}$ is a penalty on the entropy of model predictions and $\theta$ refers to the set of $\mathcal{Q}$, $\mathcal{K}$, and $\mathcal{V}$ as the model parameters. To penalize high-entropy samples, we first compute an entropy-based threshold $\eta_{ent}(x)$ as,

$$\eta_{ent}(x) = \frac{1}{\exp(H(x) - H_1)} \cdot \mathbb{1}_{\{H(x)<H_1\}} \tag{7}$$

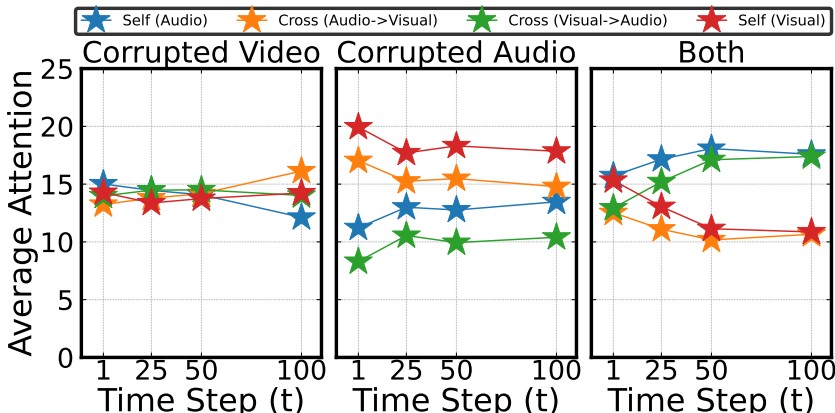

Figure 8: *Over time steps (t) during online TTA, AV2C begins to mitigate the modality bias. VGGSound [10] is audio-dominant, i.e., audio has task-specific information. AV2C begins to put more self-attention and cross-attention weights on audio.* Average attention weights are computed across 12 heads from 1 block of CAV-MAE's joint encoder for a batch size of 64. The numbers indicate averaged attention, scaled by 10,000. We show *Gaussian* on VGGSOUND-2C.

With *H(x)* being the entropy and $H_1$ being a fixed threshold, we see that high-entropy samples are penalized more and excluded from the adaptation process. Essentially, samples with low-entropy predictions are more reliable and contribute more effectively to the audio-visual TTA process. However, simply using all low-entropy samples may introduce redundancy, as similar inputs often yield similar gradients, which could hurt adaptation. To promote diversity among the selected samples, we introduce a filtering mechanism. Specifically, we maintain a running exponential moving average of the model's predicted class probabilities across recent batches, denoted as $\widehat{k}$, up to the current time step $t$. For each incoming test sample, we compute the cosine similarity between its prediction and $\widehat{k}$ to assess redundancy and retain only sufficiently dissimilar (i.e., diverse) low-entropy samples for adaptation. That is,

$$\eta_d(x) = \mathbb{1}_{\left\{\text{sim}(p(y_t|x),\widehat{k})<\rho\right\}}(x) \tag{8}$$

where, sim refers to the cosine similarity and $\rho$ is a threshold. Overall, $\widehat{\eta(x)} = \eta_{ent}(x) \cdot \eta_d(x)$.

Overall, our proposed TTA method AV2C, is a simple audio-visual TTA approach, inspired by [97, 66]. It focuses on performing on-the-fly cross-modal fusion with the $\mathcal{Q}$, $\mathcal{K}$, and $\mathcal{V}$ weights being updated based on reliable multimodal samples. Our goal is to push and give new directions for using AVROBUSTBENCH to understand and to inspire the development of more robust adaptation strategies in real-world settings. In our experiments, we set $H_1$ to $0.4\times \log(C)$, following EATA [66], and $\rho$ is set to 0.05. We update the attention parameters with a learning rate of 1e-4 for VGGSOUND-2C and 3e-4 for KINETICS-2C using the Adam update rule, with a batch size of 16.

● AV2C **minimizes modality-bias.** From Figure 8, we observe that on VGGSOUND-2C, which contains dominant task-specific audio cues [10] and with both modalities corrupted, the model gradually increases its attention to the audio tokens to perform better recognition.

## A.4   Additional Results

### A.4.1   Corruption specific results on AUDIOSET-2C, VGGSOUND-2C, KINETICS-2C, and EPICKITCHENS-2C

In Tables 9 and 10, we report the direct inference results of supervised (UAVM [29], CAV-MAE [30], EquiAV [50], TBN [48], and TIM [8]) and self-supervised models (AudioCLIP [35], ImageBind [26], and WavCLIP [95]) at a corruption-specific level/task.

Table 9: Metrics of audio-visual models evaluated on AUDIOSET-2C, VGGSOUND-2C, and KINETICS-2C at a severity level of 5. For AUDIOSET-2C, we report the mean of *MAP*, while for VGGSOUND-2C and KINETICS-2C, we report the accuracy (*Acc*).

| | Model | Gaussian | Impulse | Shot | Speckle | Compression | Snow | Frost | Spatter | Wind | Rain | Underwater | Concert | Smoke | Crowd | Interference | Mean |
|---|---|---|---|---|---|---|---|---|---|---|---|---|---|---|---|---|---|
| AUDIOSET-2C | UAVM [28] | 27.95 | 28.19 | 28.67 | 30.71 | 23.57 | 26.46 | 35.06 | 35.58 | 33.96 | 27.54 | 31.63 | 37.22 | 30.36 | 39.02 | 42.86 | 31.91 |
| | CAV-MAE [30] | 27.39 | 28.55 | 27.23 | 27.85 | 13.81 | 26.98 | 35.92 | 37.06 | 38.70 | 28.23 | 33.10 | 38.24 | 31.76 | 40.06 | 44.68 | 31.97 |
| | EquiAV [50] | – | – | – | – | – | – | – | – | – | – | – | – | – | – | – | – |
| | AudioCLIP [35] | 10.27 | 9.71 | 7.14 | 7.48 | 8.98 | 8.88 | 12.14 | 13.13 | 18.13 | 11.86 | 14.63 | 8.97 | 9.05 | 16.93 | 23.61 | 12.06 |
| | ImageBind [26] | 6.34 | 6.79 | 6.75 | 9.24 | 7.85 | 8.73 | 10.45 | 12.21 | 10.91 | 9.51 | 9.58 | 12.08 | 9.58 | 12.84 | 16.59 | 9.96 |
| | WavCLIP [95] | 0.89 | 0.91 | 0.92 | 1.62 | 1.58 | 1.17 | 1.61 | 1.85 | 2.29 | 1.05 | 1.69 | 1.55 | 1.25 | 3.38 | 4.40 | 1.74 |
| VGGSOUND-2C | UAVM [28] | 13.77 | 24.53 | 14.96 | 24.20 | 8.74 | 15.35 | 29.67 | 40.42 | 36.16 | 23.01 | 28.96 | 38.84 | 24.74 | 40.82 | 46.91 | 27.41 |
| | CAV-MAE [30] | 20.16 | 15.31 | 19.28 | 25.48 | 20.24 | 31.24 | 41.38 | 44.59 | 47.15 | 32.69 | 32.40 | 44.44 | 33.78 | 47.46 | 51.11 | 33.78 |
| | EquiAV [50] | 20.16 | 15.31 | 19.28 | 25.48 | 20.24 | 31.24 | 41.38 | 44.59 | 47.15 | 32.69 | 32.4 | 44.44 | 33.78 | 47.46 | 51.11 | 33.78 |
| | AudioCLIP [35] | 6.71 | 6.11 | 7.25 | 8.61 | 8.25 | 9.29 | 10.91 | 12.55 | 13.48 | 12.27 | 11.46 | 17.34 | 8.88 | 17.90 | 16.16 | 11.14 |
| | ImageBind [26] | 8.95 | 10.54 | 8.98 | 10.32 | 1.75 | 4.96 | 11.74 | 14.90 | 12.60 | 3.97 | 7.25 | 9.91 | 12.78 | 10.07 | 24.84 | 10.24 |
| | WavCLIP [95] | 0.56 | 0.59 | 0.59 | 3.92 | 3.13 | 1.89 | 3.92 | 5.74 | 7.38 | 1.17 | 4.72 | 4.96 | 3.63 | 12.86 | 19.82 | 4.99 |
| KINETICS-2C | UAVM [28] | 37.15 | 33.24 | 35.62 | 37.94 | 34.56 | 31.23 | 54.10 | 60.73 | 62.56 | 48.07 | 46.71 | 58.29 | 41.37 | 69.21 | 70.08 | 48.06 |
| | CAV-MAE [30] | 51.34 | 48.82 | 51.27 | 46.90 | 44.88 | 47.88 | 59.97 | 63.16 | 68.76 | 58.54 | 61.51 | 66.80 | 48.15 | 74.81 | 79.44 | 58.15 |
| | EquiAV [50] | 55.29 | 55.26 | 50.91 | 55.29 | 55.03 | 58.15 | 64.22 | 69.24 | 71.81 | 67.05 | 62.36 | 72.36 | 59.47 | 79.30 | 80.23 | 63.73 |
| | AudioCLIP [35] | 13.82 | 12.66 | 16.55 | 21.05 | 19.25 | 19.99 | 24.66 | 27.07 | 26.33 | 25.65 | 23.98 | 34.97 | 19.70 | 36.87 | 33.04 | 23.57 |
| | ImageBind [26] | 26.97 | 29.93 | 27.32 | 30.95 | 6.81 | 14.43 | 22.37 | 37.03 | 33.46 | 13.05 | 23.88 | 25.91 | 32.37 | 29.73 | 48.22 | 26.82 |
| | WavCLIP [95] | 4.50 | 4.37 | 4.85 | 15.59 | 12.21 | 8.52 | 17.49 | 21.99 | 25.72 | 7.68 | 19.09 | 20.28 | 12.83 | 38.09 | 45.55 | 17.25 |

Table 10: Metrics of TBN [48] and TIM [8] evaluated on EPICKITCHENS-2C at a severity level of 5.

| Model | Gaussian | Impulse | Shot | Speckle | Compression | Snow | Frost | Spatter | Wind | Rain | Underwater | Concert | Smoke | Crowd | Interference | Mean |
|---|---|---|---|---|---|---|---|---|---|---|---|---|---|---|---|---|
| TBN [48](Noun) | 23.32 | 22.26 | 23.46 | 16.30 | 19.91 | 21.68 | 28.05 | 25.67 | 36.05 | 26.16 | 20.42 | 27.95 | 17.22 | 35.40 | 41.32 | 25.68 |
| TBN [48](Verb) | 54.45 | 53.92 | 53.88 | 42.80 | 38.95 | 51.26 | 55.75 | 54.77 | 59.75 | 52.85 | 52.85 | 53.03 | 44.89 | 58.80 | 60.34 | 52.38 |
| TIM [8] (Noun) | 42.58 | 50.77 | 53.50 | 58.08 | 50.02 | 45.81 | 43.34 | 54.85 | 41.16 | 50.61 | 43.73 | 61.59 | 45.35 | 44.30 | 54.71 | 49.36 |
| TIM [8] (Verb) | 65.13 | 68.72 | 70.57 | 72.76 | 68.89 | 63.02 | 60.37 | 71.62 | 62.62 | 65.71 | 65.74 | 74.13 | 60.63 | 63.00 | 65.37 | 66.55 |

### A.4.2 AUDIOSET-2C and KINETICS-2C- Relative robustness for different severities

In Figures 9 and 10, we illustrate the effect of corruption severity on relative robustness on AUDIOSET-2C and KINETICS-2C. Our findings remain the same-model robustness declines with an increase in corruption severity.

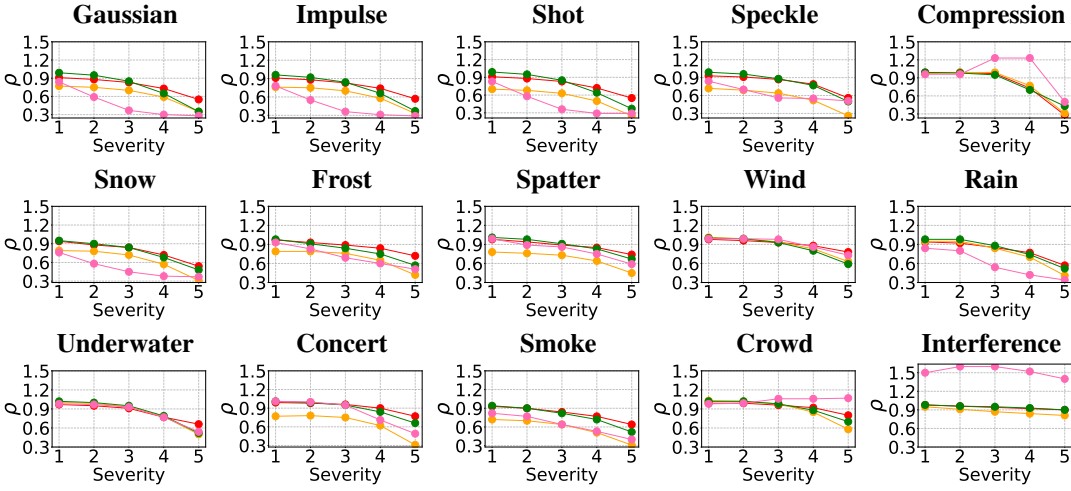

Figure 9: Relative robustness ($\rho$) on AUDIOSET-2C. We show the performance of **CAV-MAE**, **AudioCLIP**, **ImageBind**, and **Wav2CLIP**. The x-axis denotes corruption severity, and the y-axis denotes $\rho$.

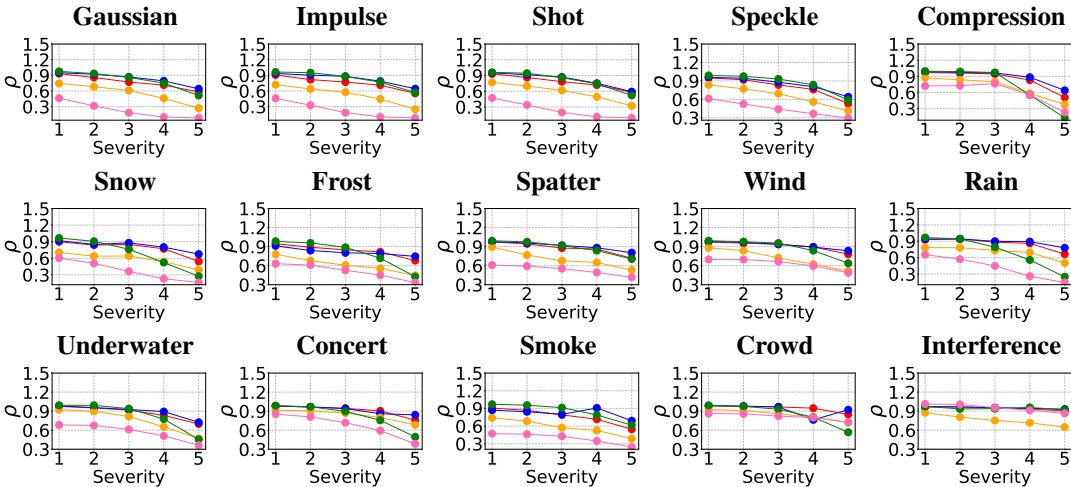

Figure 10: Relative robustness ($\rho$) on KINETICS-2C. We show the performance of **CAV-MAE**, **EquiAV**, **AudioCLIP**, **ImageBind**, and **Wav2CLIP**. The x-axis denotes corruption severity, and the y-axis denotes $\rho$.

### A.4.3 EPICKITCHENS-2C- Relative robustness for different severities

On similar lines, we illustrate the effect of corruption severity on the relative robustness of TBN [48] and TIM [8] on EPICKITCHENS-2C in Figure 11.

Table 11: *In a continual setup, with no model reset, the performance gap between mean accuracy by TTA baselines and the source model's accuracy on VGGSound (65.50%) and Kinetics-Sounds (88.10%), widens drastically.* CAV-MAE [30] is the source model initialized by VGGSound/Kinetics-Sounds weights'. We report mean accuracy (%) on VGGSOUND-2C (top) and KINETICS-2C (bottom) at a severity of 5 with a batch size of 16. Source denotes the direct inference of CAV-MAE.

| | TTA Method | Gaussian | Impulse | Shot | Speckle | Compression | Snow | Frost | Spatter | Wind | Rain | Underwater | Concert | Smoke | Crowd | Interference | Mean |
|---|---|---|---|---|---|---|---|---|---|---|---|---|---|---|---|---|---|
| VGGSOUND-2C | Source [30] | 20.39 | 23.73 | 20.72 | 25.34 | 17.26 | 25.07 | 46.82 | 48.46 | 50.17 | 29.89 | 42.19 | 47.61 | 32.93 | 47.71 | 54.88 | 35.54 |
| | TENT [90] | 1.04 | 0.33 | 0.33 | 0.36 | 0.38 | 0.33 | 0.33 | 0.33 | 0.33 | 0.33 | 0.33 | 0.33 | 0.33 | 0.33 | 0.33 | 0.38 |
| | READ [97] | 38.30 | 35.53 | 36.12 | 30.08 | 18.13 | 35.54 | 41.65 | 42.82 | 42.74 | 35.79 | 37.93 | 38.92 | 37.11 | 40.93 | 43.12 | 36.98 |
| | SuMi [34] | 22.24 | 22.90 | 21.83 | 23.29 | 16.79 | 12.97 | 44.99 | 47.30 | 48.70 | 15.74 | 40.68 | 46.46 | 28.57 | 46.57 | 54.51 | 32.90 |
| | AV2C | 38.34 | 38.35 | 39.00 | 34.52 | 21.83 | 40.06 | 47.20 | 48.63 | 49.20 | 42.29 | 44.28 | 46.66 | 44.20 | 49.70 | 50.33 | 42.31 |
| KINETICS-2C | Source [30] | 51.34 | 48.82 | 51.27 | 46.90 | 44.88 | 47.88 | 59.97 | 63.16 | 68.76 | 58.54 | 61.51 | 66.80 | 48.15 | 74.81 | 79.44 | 58.15 |
| | TENT [90] | 42.45 | 11.31 | 5.65 | 4.26 | 3.60 | 3.33 | 3.59 | 3.14 | 3.17 | 3.14 | 3.18 | 3.14 | 3.14 | 3.18 | 3.17 | 6.88 |
| | READ [97] | 53.18 | 53.01 | 54.39 | 44.25 | 37.36 | 35.21 | 43.02 | 37.18 | 33.77 | 14.27 | 20.08 | 20.05 | 13.84 | 23.88 | 24.66 | 33.85 |
| | SuMi [34] | 49.07 | 46.46 | 40.06 | 32.71 | 34.27 | 22.99 | 11.44 | 4.27 | 3.69 | 3.17 | 3.24 | 3.17 | 3.17 | 3.17 | 3.3 | 17.79 |
| | AV2C | 51.71 | 53.50 | 54.45 | 52.87 | 35.61 | 52.25 | 65.75 | 61.13 | 69.92 | 61.21 | 61.04 | 68.00 | 55.00 | 75.12 | 78.64 | 59.75 |

### A.4.4 Continual Online TTA

Online TTA focuses on adapting a pre-trained source model to a single target domain at a time. However, this assumption is often unrealistic in dynamic, real-world settings where models encounter sequences of non-stationary and evolving target domains with rapid shifts in test distributions and no knowledge of task boundaries [92]. In such a case, there are two potential challenges. The first is *catastrophic forgetting* [31]. Due to continual model parameter updates on long sequences of tasks involving unlabeled data of different distributions, there is a long-term loss of the model's source knowledge. The second challenge is *error accumulation*. As updates happen on noisy test data, errors in early adaptation steps can propagate and compound over time, leading to significant degradation in performance.

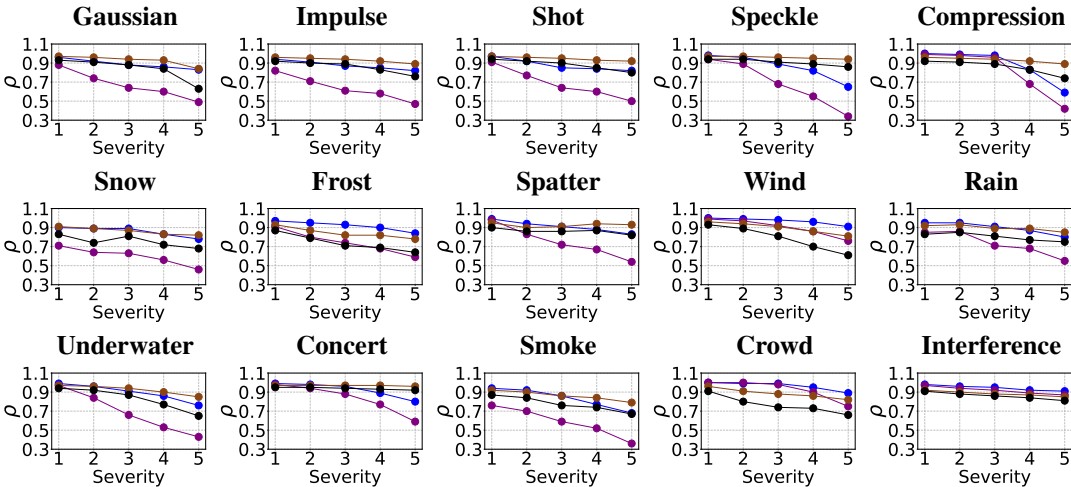

Figure 11: Relative robustness ($\rho$) on Epic-Kitchens-2C (EPICKITCHENS-2C). We show **TBN (Noun)**, **TBN (Verb)**, **TIM (Noun)**, and **TIM (Verb)**. The x-axis denotes corruption severity; the y-axis denotes $\rho$.

In this section, we extend our study of online TTA to a continual setting where the CAV-MAE source model is not reset at any point in time or after any domain, and being continually fine-tuned to the tasks. We present the results in Table 11. Experiments are performed on VGGSOUND-2C and KINETICS-2C.

## A.5 Subjective Evaluations - Humans are very effective in recognizing corrupted audios and videos

**Motivation.** Geirhos et al. [21] demonstrate a notable gap between human and model robustness on noisy images. From an audio-visual standpoint, humans naturally integrate cross-modal cues to interpret and learn from their surroundings [82]. We bridge the two ideas to study this from a subjective point of view.

**Setup and Participants.** We recruited 30 volunteers from diverse backgrounds, with most participants falling within the 18–50 age range. This recognition study aimed to evaluate the effectiveness of AVROBUSTBENCH and investigate human performance under severe audio and visual distributional shifts. The central question we sought to answer was: Can humans still reliably recognize when both modalities are corrupted? If so, this underscores the importance of developing models that are not only robust but also adaptive to the challenges of an open and dynamic world.

To begin our experiment, we designed a user-friendly interface. From the VGGSOUND-2C dataset, we manually sampled 30 challenging videos with each featuring overlaid corrupted audio and drawn from any of the 15 proposed audio-visual corruptions at a severity level 5. We choose a small subset to not overburden a participant. For each instance, each participant was shown a corrupted video alongside a set of 20 plausible labels and asked to select the one that best described the action depicted. We did not show all the 309 labels since that would have made it even more challenging. Each participant, on average, took about 20-30 seconds to identify the action in a video.

**Results.** We discuss the results here. Averaged across all the participants, the reported human accuracy was ∼89%. Qualitatively, we observed that participants found *Digital*-ly corrupted videos slightly difficult to recognize. The participants did mention that, depending on the video, they relied on the audio or visual cues to identify an action. Likely, since there is a pixel-level and frequency-level disturbance, this hindered human recognition. Similar to our findings, videos with *Human-related* corruptions were very easy to identify. These corruption types are often observed and are familiar by humans. The major takeaway from these experiments is that human perception remains robust under many real-world corruptions. This highlights the importance of designing audio-visual models that can similarly adapt to and withstand such conditions in open-world environments.

## A.6 Audio-Visual Retrieval - Cross-modal correspondence is hampered drastically

Table 12: *Models struggle to maintain cross-modal correspondence under AV corruptions at test-time.* Numbers report retrieval recalls (R@1, R@5, R@10) for Visual→Audio (left) and Audio→Visual (right) on AUDIOSET-2C and VGGSOUND-2C subsets. We report the mean metrics across the proposed 15 tasks, computed at a severity of 5. "Clean" refers to the original test sets. We use CAV-MAE [30] as the pre-trained model.

| Visual→Audio | AUDIOSET-2C | | | VGGSOUND-2C | | |
|---|---|---|---|---|---|---|
| | R@1 | R@5 | R@10 | R@1 | R@5 | R@10 |
| Clean | 16.63 | 35.71 | 45.15 | 12.62 | 28.48 | 37.00 |
| Across 15 tasks | 0.97 | 2.94 | 3.33 | 1.51 | 4.57 | 6.60 |

| Audio→Visual | AUDIOSET-2C | | | VGGSOUND-2C | | |
|---|---|---|---|---|---|---|
| | R@1 | R@5 | R@10 | R@1 | R@5 | R@10 |
| Clean | 13.41 | 29.42 | 38.36 | 12.76 | 28.43 | 36.36 |
| Across 15 tasks | 0.91 | 2.69 | 4.06 | 2.24 | 6.72 | 10.24 |

While CAV-MAE [30] claims to learn rich joint audio-visual representations, we now investigate whether such a supervised pre-trained model can effectively capture audio-visual correspondences under real-world distributional shifts at test-time. Here, we study audio-visual retrieval, which relies on semantic alignment between audio and visual content for cross-modal search. Following the setup from CAV-MAE, we uniformly sample pairs from AUDIOSET-2C and VGGSOUND-2C, creating subsets of 1,725 and 1,545 samples, respectively. Retrieval performance is evaluated using cosine similarity between the modality representations and reported as retrieval recall at ranks 1, 5, and 10. The results of audio→visual and visual→audio are reported in Table 12. Given a corrupted query modality, we retrieve the other modality. We report metrics on a clean subset, which may slightly differ from the original CAV-MAE due to variations in the test subsets and YouTube URL availability. However, the main takeaway lies in the large recall gap between clean subsets and the average performance. On AUDIOSET-2C and VGGSOUND-2C, R@1 drops by 15.66% and 12.5% respectively.

Table 13: Metrics of audio-visual LLMs evaluated on VGGSOUND-2C and KINETICS-2C at a severity level of 5 for action recognition. We report the accuracy (*Acc*) on each task. For comparison, we also provide the performances on the clean/original test sets.

| | Model | Gaussian | Impulse | Shot | Speckle | Compression | Snow | Frost | Spatter | Wind | Rain | Underwater | Concert | Smoke | Crowd | Interference | Clean |
|---|---|---|---|---|---|---|---|---|---|---|---|---|---|---|---|---|---|
| VGGSOUND-2C | VideoLLaMA-2 [12] | 20.73 | 38.49 | 14.81 | 42.70 | 39.91 | 18.78 | 28.93 | 38.89 | 39.62 | 24.55 | 30.03 | 43.52 | 25.41 | 50.01 | 50.99 | 55.80 |
| | PandaGPT [83] | 3.90 | 7.00 | 3.09 | 7.71 | 6.85 | 2.04 | 5.49 | 5.39 | 4.91 | 3.18 | 2.68 | 5.61 | 3.61 | 7.50 | 9.13 | 11.87 |
| KINETICS-2C | VideoLLaMA-2 [12] | 21.67 | 24.94 | 24.76 | 42.91 | 53.84 | 48.41 | 46.77 | 60.95 | 42.78 | 55.35 | 44.81 | 66.44 | 18.00 | 66.83 | 69.78 | 76.37 |
| | PandaGPT [83] | 6.94 | 7.68 | 7.3 | 10.67 | 9.19 | 6.36 | 8.16 | 10.67 | 9.1 | 6.27 | 6.72 | 15.30 | 7.20 | 13.28 | 16.19 | 22.24 |

## A.7 Robustness of Audio-Visual LLMs

Given the success of Multimodal Large Language Models (MLLMs) [98] in various understanding tasks, we touch upon and explore their robustness on our proposed audio-visual datasets. Specifically, we use Audio-Visual LLMs (AVLLMs) i.e. VideoLLaMA-2 [12] and PandaGPT [83] for the audio-visual recognition task on VGGSOUND-2C and KINETICS-2C.

The evaluation approach of these multimodal LLMs differs slightly from the *supervised* and *self-supervised* models discussed in the main paper. These MLLMs take audio-visual and a text query input. For example, we prompt the model with : "Which class of VGGSound does this video belong to?" The model generates a textual output, which we compare against class labels using cosine similarity. To do this, we encode both the predicted output and the class labels using the CLIP text encoder and compute similarity scores. The highest similarity label is considered the predicted label for this specific audio-visual pair.

Since KINETICS-2C has 32 labels, we use the following prompt during inference with MLLMs: "*Which of the following actions best describe the content of this video? Choose one from the list below: [labels].* " The model generate a textual response as output. Similarly, we use the text encoder in CLIP to compute the cosine similarity between predicted and ground-truth labels in the embedding space to calculate the accuracy.

The results are shown in Table 13. Consistent with findings from supervised and self-supervised models, we observe AVLLMs show large performance degradation under audio and visual distributional shifts. While effective prompting techniques or other strategies can be explored, we leave that for future work.

