# OpenReview forum: "$\texttt{AVROBUSTBENCH}$: Benchmarking the Robustness of Audio-Visual Recognition Models at Test-Time"
_NeurIPS.cc/2025/Datasets_and_Benchmarks_Track — NeurIPS 2025 Datasets and Benchmarks Track poster_

### Official Review · Reviewer_kPFY · 2025-06-05

**Rating:** 4
**Confidence:** 2

**Summary:**

The paper introduces AVROBUSTBENCH, a benchmark that is designed to evaluate the robustness of audio-visual recognitiion models against corruptions at test-time. The proposed benchmarkcomprises four audio-visual benchmark datasets, AUDIOSET-2C,VGGSOUND-2C,KINETICS-2C,and EPICKITCHENS-2C, each incorporating 75 bimodal audio-visual corruptions that are co-occurring and correlated. The paper observe that the SOTA supervised and self-supervised audio-visual models all exhibit declining robustness as corruption severity increases. Along with the dataset, the paper also proposed AV2C, an online test-time adaptation (TTA) method designed to improve the robustness audio-visual models under the corruption.

**Dataset Code Accessibility:**

Yes

**Ethical Considerations:**

No, there are no or only very minor ethics concerns

**Final Justification:**

I appreciate the clarification and the effort the author put into addressing the reviewers' concerns. Hence, I have increased my final rating.

**Limitations Weaknesses:**

- The writing about AV2C is somehow unclear. It is unclear what is the novelty of the proposed AV2C method compare with the previous method.

- Although the author provide extensive evaluation results on four proposed benchmark, it is unclear how will the model act and perform on the wider range of audio-visual learning task (i.e., audio-visual segmentation [a], source-sound seperation [b]). It is essential to present results on these benchmarks.

[a] Zhou, J., Wang, J., Zhang, J., Sun, W., Zhang, J., Birchfield, S., Guo, D., Kong, L., Wang, M. and Zhong, Y., 2022, October. Audio–visual segmentation. In European Conference on Computer Vision (pp. 386-403). Cham: Springer Nature Switzerland.

[b] Gao, R. and Grauman, K., 2019. Co-separating sounds of visual objects. In Proceedings of the IEEE/CVF International Conference on Computer Vision (pp. 3879-3888).

**Strengths Contributions:**

- The paper is generally well-written and easy to follow.
- The idea of evaluating robustness of audio-visual recognition models is interesting and important for the audio-visual community.
- The paper perform extensive evaluation on four new benchmarks.

---

> ### Author Rebuttal · Authors · 2025-07-31
>
> We thank Reviewer kPFY for finding our work well-written, easy to follow, and interesting. Below, we address the questions and concerns.
>
>
> 1. Novelty of $\texttt{AV2C}$: We apologize for not being very clear! To clarify, we identified that existing audio-visual TTA methods drastically underperform in the presence of bimodal corruptions. To exchange cross-modal information on-the-fly, we adapt the QKV attention parameters of the joint-encoder for reliable fusion via modeling the self- and cross-attention, at test-time, following READ [1]. However, under such severe noise on both modalities, simple entropy minimization is not a good idea since high-entropy samples can degrade the performance. So, the parameters are updated based on the low-entropy samples, by weighing them more, to enhance model predictions. This is based on EATA [2]. Such a simple lightweight approach beat all the TTA baselines on our benchmark. To understand the need for QKV adaptation by ablation, we adapt LayerNorm (LN) parameters of all encoders, and the model classifier (CLS) on VGGSound-2C and present the results below. Interestingly, just the classifier adaptation with entropy filtering beats most baselines (see the main paper). To be clear, $\texttt{AV2C}$ is not the main goal of this work but gives possible directions for future work on our challenging benchmark.
>
> | Params.     | Gaussian | Impulse | Shot   | Speckle | Compression | Snow  | Frost | Spatter | Wind  | Rain  | Underwater | Concert | Smoke | Crowd | Interference | **Mean** |
> |------------|----------|---------|--------|---------|-------------|-------|-------|---------|-------|-------|------------|---------|-------|-------|--------------|----------|
> | QKV   | 38.27   | 37.14   | 37.43  | 32.45   | 21.18       | 40.78 | 50.13 | 52.33   | 53.60 | 43.98 | 46.51     | 51.10   | 46.74 | 53.9 | 54.84        | 44.03
> | LN     | 37.20   | 36.53   | 36.71  | 34.89   | 25.60       | 38.42 | 49.28 | 50.80   | 51.76 | 42.38 | 46.87     | 50.39   | 37.05 | 52.36 | 54.54        | 42.98
> | CLS    | 28.87   | 29.26   | 29.14  | 28.16  | 16.26      | 29.95 | 48.41 | 49.44  | 51.29 | 36.4 | 43.15     | 48.83   | 36.34 | 50.02 | 54.76        | 38.69 |
>
>
> 2. Robustness studies for other tasks: Excellent question! We would like to point out that in the Appendix, we present audio-visual retrieval results where we show that audio-visual correspondences are drastcially hampered, resulting in low recall scores. Beyond recognition and retrieval, we perform,
>
>     a. **Audio-visual segmentation (AVS)**: We employ the SOTA model SAMA-AVS [3] and introduce our proposed corruptions (severity=5) to both the modalities of the AVSBench-S4 [6] test set, with 740 videos in each task. We report the mean IoU (mIoU) and F-score in Table 1. For comparison, the mIoU and F-score on the clean S4 are 81.53 and 0.886 respectively. **To conclude, SOTA AVS models struggle in the presence of bimodal corruptions.**
>
>     b. **Source-sound separation**: We employ the SOTA model DAVIS [4] and introduce our proposed corruptions (severity=5) to both the modalities of the MUSIC [5] test set, with 250 videos. As metrics, we report Signal to Distortion Ratio (SDR), Signal to Interference Ratio (SIR) and Signal to Artifact Ratio (SAR) in Table 2 (higher the better). For comparison, the SDR/SIR/SAR on the clean MUSIC test set are 11.68/18.36/15.26 and the mean noisy scores are 0.18/6.77/7.46 respectively. **To conclude, SOTA AV-Sound separation models struggle in the presence of bimodal corruptions.**
>
> Table 1: Audio-Visual Segmentation results.
> | Metric     | Gaussian | Impulse | Shot   | Speckle | Compression | Snow  | Frost | Spatter | Wind  | Rain  | Underwater | Concert | Smoke | Crowd | Interference | **Mean** |
> |------------|----------|---------|--------|---------|-------------|-------|-------|---------|-------|-------|------------|---------|-------|-------|--------------|----------|
> | mIoU      | 30.71    | 40.83   | 34.14  | 48.66   | 37.66       | 51.26 | 51.77 | 68.79   | 0.02 | 61.67 | 35.31      | 69.84   | 40.39 | 47.91 | 48.15       | 44.47 |
> | F-score    | 0.42    | 0.52   | 0.46  | 0.59  | 0.51      | 0.63 | 0.62 | 0.78   | 0.002 | 0.72 | 0.45      | 0.79   | 0.53 | 0.60 | 0.61        | 0.55|
>
>
>
> Table 2: Sound-source separation results.
> | Metric | Gaussian | Impulse | Shot   | Speckle | Compression | Snow   | Frost  | Spatter | Wind   | Rain   | Underwater | Concert | Smoke  | Crowd  | Interference | **Mean** |
> |--------|----------|---------|--------|---------|-------------|--------|--------|---------|--------|--------|------------|---------|--------|--------|--------------|--------------|
> | SDR    | -0.03   | -1.95  | -1.65 | -0.34  | 5.45       | -1.00 | -0.44 | 0.36   | -1.62  | -1.45 | -2.72     | 0.017  | -1.11 | -1.3  | 10.58       | 0.18       |
> | SIR    | 5.96    | 4.04    | 4.39  | 4.39   | 12.48      | 4.48  | 6.43  | 7.54   | 4.93  | 4.41  | 4.85      | 6.46   | 4.46  | 7.15  | 19.57       | 6.77       |
> | SAR    | 8.43    | 6.44   | 6.61  | 6.61   | 11.11      | 7.13   | 5.97  | 7.89   | 7.35  | 6.68  | 3.87      | 7.01   | 8.17  | 3.96  | 14.72        | 7.46        |
>
>
>
> [1] Yang, M., Li, Y., Zhang, C., Hu, P., Peng, X.: Test-time adaptation against multi-modal reliability bias. In: The Twelfth International Conference on Learning Representations (2024)
>
> [2] Niu, S., Wu, J., Zhang, Y., Chen, Y., Zheng, S., Zhao, P., Tan, M.: Efficient test-time model adaptation without forgetting. In: International conference on machine learning. pp. 16888–53616905. PMLR (2022)
>
> [3] Liu, J., Wang, Y., Ju, C., Ma, C., Zhang, Y., Xie, W.: Annotation-free audio-visual segmentation. In: Proceedings of the IEEE/CVF Winter Conference on Applications of Computer Vision (2024)
>
> [4] Huang, C. and Liang, S. and Tian, Y. and Kumar, A. and Xu, C.: DAVIS: High-Quality Audio-Visual Separation with Generative Diffusion Models. In: Proceedings of the Asian Conference on Computer Vision (ACCV) (2024)
>
> [5] Hang Z., Chuang G., Andrew R., Carl V., Josh M., Antonio T.: The Sound of Pixels. In: Proceedings of the European Conference on Computer Vision (ECCV), 2018
>
> [6] Zhou, J., Wang, J., Zhang, J., Sun, W., Zhang, J., Birchfield, S., Guo, D., Kong, L., Wang, M., Zhong, Y.: Audio-visual segmentation.
> In: European Conference on Computer Vision (2022)

---

### Official Review · Reviewer_sppH · 2025-06-20

**Rating:** 5
**Confidence:** 3

**Summary:**

This work introduces AVROBUSTBENCH, a comprehensive benchmark designed to evaluate the test-time robustness of audio-visual recognition models. AVROBUSTBENCH comprises four audio-visual benchmark datasets, AUDIOSET-2C , VGGSOUND-2C , KINETICS-2C , and EPICKITCHENS-2C , each incorporating 75 bimodal audio-visual corruptions that are co-occurring and correlated.

**Dataset Code Accessibility:**

Yes

**Dataset Code Comments:**

The dataset is available on the HF platform, which is simple and easy to access

**Ethical Considerations:**

No, there are no or only very minor ethics concerns

**Limitations Weaknesses:**

1. Compared to the AUDIOSET-2C dataset, why is the performance degradation more pronounced on other datasets?

**Strengths Contributions:**

1. The overall structure of the paper is good and very informative.
2. The results of the experiments were very convincing and the dataset was reviewed in its entirety.

---

> ### Author Rebuttal · Authors · 2025-07-31
>
> We thank Reviewer sppH for finding our work informative. Below, we address the concern.
>
> 1. Performance drop on other datasets: Thank you for this insightful question! We believe the drop on AudioSet‑2C is noticeably smaller for two main reasons,
>
>     a. **Evaluation metrics**: AudioSet, thereby AudioSet-2C, is multi‑label and evaluated with MAP (mean average precision), a ranking‑based metric where small score shifts rarely change the label ordering, so drops appear moderate. VGGSound-2C and Kinetics-2C are single‑label and use top‑1 accuracy, where even a slight score change that lowers the correct class from first to second counts as a full error, making the drop look much larger.
>
>     b. **Pre-training data characteristics**: AudioSet is comparatively large (about 2M samples) and inherently noisy, since its clips come from unconstrained YouTube videos with overlapping sounds and background noise. Models trained on this noisy data are, in effect, already exposed to many kinds of natural corruptions, which makes them more resilient to our proposed audio‑visual corruptions added at test-time. On the other hand, VGGSound and Kinetics-Sounds are relatively smaller (about 200k and about 22k respectively), more carefully filtered datasets with cleaner labels and clips. Because models trained on these datasets have had less exposure to noisy or corrupted conditions, their performance drops more sharply when our unseen corruptions are introduced.

---

### Official Review · Reviewer_bpLn · 2025-07-03

**Rating:** 5
**Confidence:** 3

**Summary:**

This paper introduces AVROBUSTBENCH, a comprehensive benchmark for evaluating the test-time robustness of audio-visual recognition models. Unlike prior benchmarks that focus on unimodal or disjoint corruptions, AVROBUSTBENCH introduces 75 bimodal, co-occurring, and correlated corruption types applied across four datasets: AUDIOSET-2C, VGGSOUND-2C, KINETICS-2C, and EPICKITCHENS-2C. The authors evaluate both supervised (e.g., UAVM, CAV-MAE, TIM) and self-supervised (e.g., AudioCLIP, ImageBind) models under increasing corruption severity and observe substantial performance degradation. In response to the limitations of existing online test-time adaptation (TTA) methods, the authors also propose AV2C, a TTA approach that selectively adapts QKV attention weights using low-entropy samples for robust cross-modal fusion.

**Dataset Code Accessibility:**

Yes

**Ethical Considerations:**

No, there are no or only very minor ethics concerns

**Final Justification:**

The authors have addressed my concerns in the rebuttals, I'm satisfied with the submission and have no further concerns. More specifically, the authors have addressed my concerns regarding:
- Novelty of AV2C
-  Benchmark Overlap with Prior Work

**Limitations Weaknesses:**

## Limited Novelty of AV2C

While AV2C performs well empirically, it appears to be a marginal extension of existing TTA methods, particularly:
- EATA, which uses entropy minimization to guide adaptation.
- READ, which adapts QKV weights in the joint encoder for cross-modal robustness.

AV2C essentially fuses these two ideas without introducing a significantly new mechanism or theoretical perspective. There is no formal analysis, and no principled explanation of why the combination of these techniques leads to improved robustness under bimodal corruption.

To increase the impact of AV2C, the paper would benefit from any of these analyses:
- Ablation studies separating the contributions of entropy filtering and QKV adaptation.
- Visualization of entropy distributions to justify sample selection with QKV adaptation.
- Case studies or failure mode analysis to show where AV2C succeeds and others fail.
- Analytical insight into how attention dynamics shift under bimodal noise and why AV2C’s updates are more effective than those in existing approaches.

## Benchmark Overlap with Prior Work

At the benchmark level, while AVROBUSTBENCH introduces valuable innovations such as co-occurring, correlated, and multimodal corruptions, it’s important to acknowledge other similarities and differences with existing benchmarks (as shown in table 1). For example, the READ paper also introduced VGGSound-C and Kinetics-50C, which apply analogous corruption types (e.g., Gaussian noise, impulse noise) and severity levels to individual modalities.

Authors could elaborate more on how AVROBUSTBENCH (potentially in table 1) has broader and more diverse set of tasks across four datasets (AUDIOSET-2C, VGGSOUND-2C, KINETICS-2C, EPICKITCHENS-2C). These differences do meaningfully elevate the benchmark’s scope A side-by-side visual or metric comparison to VGGSound-C and Kinetics-50C (in terms of corruption diversity, task coverage, or label space) would help reinforce AVROBUSTBENCH’s added value.

## Minor Comments

- Typo: “co-occuring” → “co-occurring” (appears multiple times).
- Inconsistent Acronym Usage:
1. Some acronyms (e.g., Discrete Cosine Transform (DCT)) are introduced but not reused.
2. Others, like test-time adaptation (TTA), are redundantly defined. I suggest performing a pass for acronym consistency.

**Strengths Contributions:**

The paper is generally well-written, logically organized, and easy to follow.

Key strengths include:
- Novel Benchmark for Bimodal Corruptions: AVROBUSTBENCH realistically models real-world distributional shifts by introducing co-occurring, correlated audio-visual corruptions that challenge model robustness more severely than prior unimodal or disjoint benchmarks.
- Broad Dataset Coverage: The benchmark spans four well-established AV datasets, each enriched with corruption types across Digital, Environmental, and Human-Related categories, offering diverse and representative test scenarios.
- Comprehensive Robustness Evaluation: The study provides detailed performance breakdowns across corruption types and severity levels, revealing that both supervised and self-supervised models struggle significantly under noise.
- Critical Analysis of TTA Methods: Existing TTA baselines such as TENT, EATA, READ, and SuMi are evaluated. The results show that these methods are often ineffective under bimodal corruption, particularly for Digital distortions.
- Proposed AV2C Method: AV2C combines entropy-based sample selection with QKV attention adaptation, achieving consistent improvements on VGGSOUND-2C and competitive results on KINETICS-2C.

---

> ### Author Rebuttal · Authors · 2025-07-31
>
> We thank Reviewer bpLn for finding our paper to be well-written, logically organized, and an easy-read. We address the questions and concerns below.
>
> 1. Limited novelty of $\texttt{AV2C}$: Thank you for raising this concern.
>
>     a. While $\texttt{AV2C}$ builds on encouraging directions from EATA and READ, we show that existing TTA methods perform poorly on our benchmark. As shown in Table 2 below, even a simple adaptation of the classifier based on entropy filtering outperforms most existing TTA baselines. But, substantial gains come from dynamic cross-modal fusion via QKV adaptation with entropy filtering. $\texttt{AV2C}$ serves as a cleverly designed, lightweight approach that addresses this. In Appendix A.3, we provide fine-grained details of $\texttt{AV2C}$ and show in Figure 2 (Appendix) that $\texttt{AV2C}$ helps alleviate the modality bias existing in READ. Our goal was to provide possible directions for future TTA research on our challenging benchmark to make methods more robust.
>     b. However, the major contribution of our work is the proposal of the datasets in $\texttt{AVROBUSTBENCH}$ and our analysis on limited robustness of SOTA audio-visual models. We also shed light on the limitedness of existing TTA methods.
>
> 2. Further experimental suggestions for $\texttt{AV2C}$: Thank you for the suggestions! We show ablation results on VGGSound-2C here.
>
>     a. We ablate the entropy threshold $H_1$ (originally 0.4xlogC, C = #classes) that weights low-entropy samples more. The results are in Table 1. We see that higher thresholds give better or comparable performances suggesting that, under severe bimodal noise, filtering out high-entropy samples is indeed necessary. A threshold of 0.2xlogC yields better scores than READ and SuMi too (see main paper). We also ablate the need for QKV adaptation in Table 2. We adapt the LayerNorm (LN) layers of all the encoders, and the model classifier (CLS) only. Results show the need for dynamic cross-modal fusion with entropy filtering. Interestingly, on average, just CLS adaptation betters most TTA baselines including READ (see Table 6 of main paper), highlighting the need for entropy filtering.
>
>    b. We will add the suggested visualizations and other case studies in the revised draft.
>
>    c. For attention dynamics, please check the Appendix (Figure 2). Our method addresses the modality bias happening in READ [2].
>
> Table 1: Ablation on the entropy threshold $H_1$.
> | $H_1$     | Gaussian | Impulse | Shot   | Speckle | Compression | Snow  | Frost | Spatter | Wind  | Rain  | Underwater | Concert | Smoke | Crowd | Interference | **Mean** |
> |------------|----------|---------|--------|---------|-------------|-------|-------|---------|-------|-------|------------|---------|-------|-------|--------------|----------|
> | 0.2xlogC    | 38.4   | 36.99  | 37.98   | 33.75    | 19.03       | 40.12 | 50.1  | 51.86    | 52.94 | 43.85  | 46.67      | 51.18   |45.15  | 53.58 |    54.73      | 43.76 |
> | 0.4xlogC     | 38.27   | 37.14   | 37.43  | 32.45   | 21.18       | 40.78 | 50.13 | 52.33   | 53.60 | 43.98 | 46.51     | 51.10   | 46.74 | 53.9 | 54.84        | 44.03
> | 0.6xlogC    | 38.76     | 38.0    | 38.13   | 34.74    | 21.67       | 40.99  | 50.28  | 52.38    | 53.06  | 44.32  | 46.54       | 51.37    |46.74  | 53.87 |    54.94      | 44.39 |
> | 0.8xlogC    |  38.53  |  37.82  |  38.1  | 34.77    |   21.95    | 40.99  | 50.23 |  52.31   | 53.2  |44.25  | 46.55       |  51.43   | 46.35 | 53.74 |   54.8     | 44.33 |
>
>
> Table 2: Ablation of the choice of parameters for TTA with fixed entropy filtering.
> | Params.     | Gaussian | Impulse | Shot   | Speckle | Compression | Snow  | Frost | Spatter | Wind  | Rain  | Underwater | Concert | Smoke | Crowd | Interference | **Mean** |
> |------------|----------|---------|--------|---------|-------------|-------|-------|---------|-------|-------|------------|---------|-------|-------|--------------|----------|
> | QKV   | 38.27   | 37.14   | 37.43  | 32.45   | 21.18       | 40.78 | 50.13 | 52.33   | 53.60 | 43.98 | 46.51     | 51.10   | 46.74 | 53.9 | 54.84        | 44.03
> | LN     | 37.20   | 36.53   | 36.71  | 34.89   | 25.60       | 38.42 | 49.28 | 50.80   | 51.76 | 42.38 | 46.87     | 50.39   | 37.05 | 52.36 | 54.54        | 42.98
> | CLS    | 28.87   | 29.26   | 29.14  | 28.16  | 16.26      | 29.95 | 48.41 | 49.44  | 51.29 | 36.4 | 43.15     | 48.83   | 36.34 | 50.02 | 54.76        | 38.69 |
>
>
>
>
> 3. Overlap with VGGSound-C and Kinetics-50C: Thank you for pointing this out! We have discussed this minimally in the main paper (Table 1 and Related Works) due to a lack of space.
>
>     a. **Differences**: Majorly, VGGSound-C and Kinetics-50C have unimodal audio and visual corruptions. Each dataset has 6 audio and 15 visual corruptions, respectively, and these are disjoint and unrelated across modalities. Instead, our work is inspired by a high possibility of real-world shifts where both the modalities are corrupted yet correlated, caused due to shared environmental conditions. We also propose a wider and broader set of corruptions ranging from digital, weather, to human-related bimodal shifts. Notably, the visual corruptions in VGGSound-C and Kinetics-50C are directly taken from ImageNet-C [1], while the audio corruptions are from recorded samples available on FreeSound. There is no cross-modal coherence.
>
>     b. **Similarities**: Like VGGSound-C and Kinetic-50C, our work too has 5 severity levels of corruptions. On a dataset level, VGGSound-C and VGGSound-2C share the same label space.
>
>
> 4. Discussions on diversity across the datasets: Great suggestion! To add on to Table 1 of our submission, we've contrasted against other benchmarks below. We'll add this to our revised draft.
>
>
> | Benchmark | #Datasets | #Corruptions | Task Diversity | Evaluation Metrics |
> |-----------|-----------|--------------|----------------|--------------------
> | READ (VGGSound-C, Kinetics-50C) [2]    |  2   | 15 (visual) + 6 (audio), 5 severities      |  Single-label action recognition, event classification        |   Accuracy       |
> | $\texttt{AVROBUSTBENCH}$ (Ours)    |  4   | 15 bimodal audio-visual, 5 severities      |  Single-label, Multi-label, Action, Retrieval       |   Accuracy, Mean Average Precision, Recall@K       |
>
>
> 5. Other minor comments: Thank you! We'll make the required changes in the revised draft.
>
>
>
> [1] Hendrycks, D., Dietterich, T.: Benchmarking neural network robustness to common corruptions and perturbations. arXiv preprint arXiv:1903.12261 (2019)
>
> [2] Yang, M., Li, Y., Zhang, C., Hu, P., Peng, X.: Test-time adaptation against multi-modal reliability bias. In: The Twelfth International Conference on Learning Representations (2024)

---

> > ### Comment · Reviewer_bpLn · 2025-08-05
> >
> > Thank you to the authors for their detailed and constructive rebuttal, and for providing further ablation studies and clarifications. I will increase my score to reflect the improvements.

---

> > > ### Author Response · Authors · 2025-08-05
> > >
> > > Really appreciate you for taking the time out to read our rebuttal.

---

### Official Review · Reviewer_5mV3 · 2025-07-03

**Rating:** 5
**Confidence:** 4

**Summary:**

This paper introduces AVROBUSTBENCH, a new benchmark designed to measure audio-visual model robustness. Unlike previous benchmarks focusing on single-modal corruption, AVROBUSTBENCH explores scenarios where both audio and visual modalities are corrupted concurrently and the corruptions are correlated. It simulates real-world digital, environmental, and human-related corruptions, creating four benchmark datasets: AUDIOSET-2C, VGGSOUND-2C, KINETICS-2C, and EPICKITCHENS-2C. The authors conduct extensive experiments evaluating various supervised and self-supervised audio-visual models on this benchmark. Furthermore, they analyze limitations of prior TTA methods for dual-modal corruption and propose a new TTA approach called AV2C.

**Dataset Code Accessibility:**

Yes

**Dataset Code Comments:**

I have checked the dataset code and download links.

**Ethical Considerations:**

No, there are no or only very minor ethics concerns

**Final Justification:**

The authors have addressed my concerns, and I have no further questions.

**Limitations Weaknesses:**

1. The authors' claim of "real-world shift" for the proposed corruptions is inaccurate, as the audio and video corruptions are artificially constructed simulations and differ from genuine distribution shifts.
2. The construction of corruption in the main text is crucial and should be detailed. The paper categorizes the 15 corruptions into three types to cover more diverse scenarios. However, the rationale and justification for this classification seem to be lacking.
3. The experiment analyzes the reasons for different models' robustness based on evaluation metrics. It is better to provide more visualization results (e.g., feature distribution) to show which model type is more robust. Also, factors affecting model robustness can be analyzed from more angles, such as model aspects (structure, size) and data aspects (pre-training data, training strategy, corruption features).
4. While the study focuses on concurrent corruption of both modalities, a comparison with single-modality corruption cases would provide additional valuable insights about model robustness.

**Strengths Contributions:**

1. AVROBUSTBENCH is the first benchmark featuring co-occurring, correlated audio-visual corruptions for robustness evaluation.
2. This paper is well-motivated and clearly written.
3. The authors provide a comprehensive experimental analysis of model robustness and TTA performance.
4. This paper release the benchmark datasets and construction code, ensuring easy accessibility and potential migration to other datasets.

---

> ### Author Rebuttal · Authors · 2025-07-31
>
> We thank Reviewer 5msV3 for finding our work well-motivated, comprehensive, and clearly written. Below, we address the questions and concerns.
>
>
> 1. Claim of "real-world shift": Thank you for pointing this out. We clarify this below.
>
>     a. While our corruptions are simulated to mimic genuine distributional shifts, our idea of "real-world shift" is the *expectation* or the *possible* occurances of **correlated** shifts that can happen in the real-world, **simultaneously** affecting both the modalities.
>
>     b. To add on to this, audio corruptions under *Environmental* and *Human-Related* are directly sampled from recorded audio, available on Freesound, and overlayed. Visual corruptions are simulated but emulate a real-world experience. Please see Appendix A.1.2 for more information. However, we will make this very clear in the revised draft.
>
> 2. Rationale behind corruption construction: Thank you for your comment. Due to a lack of space in the main text, we will clarify our rationale here.
>
>     a. Under *Digital*, we place *Gaussian*, *Impulse*, *Shot*, *Speckle*, and *Compression*. We borrow this nomenclature from ImageNet-C [1], with a slight abuse of grouping. To add on, these noises are synthetically added perturbations to visual frames and audios.
>
>     b. Corruptions under *Environmental* are grouped based on ImageNet-C, as they simulate real-world or natural effects in the wild. Our corruptions in *Human-related* portray disturbances caused due to human activites, and hence, the group name.
>
> 3. Further analysis of model robustness: Thank you for the suggestion! We further analyzed model robustness across four factors i.e. training strategy, model size, architecture, and augmentation. We report the average ρ (relative robustness) on the VGGSound-2C dataset.
>
>     a. **Training strategy**: Please see Table 1. Supervised models show higher robustness compared to self-supervised ones. This indicates that task-specific supervision provides stronger alignment for robustness.
>
>     b. **Model size**: Please see Table 2. While the models vary significantly in parameter count (from 134M to 1.2B), ρ does not consistently improve with size. Thus, robustness appears to depend more on training choices than raw parameter count.
>
>     c. **Architecture**: Please see Table 3. Transformer-based models achieve higher robustness than ResNet-based models. This is likely because their scalable representation learning and patch-wise processing make them better at handling noise compared to convolutional architectures.
>
>     d. **Augmentations**: Please see Table 4. We compare models with and without training augmentations and find that those with augmentations are more robust, likely because exposure to varied inputs improves generalization to noisy or unseen data.
>
> Table 1: Training strategy vs $\rho$
> |  | Models | $\rho$|
> |----------|----------|----------|
> | Supervised  | UAVM, CAV-MAE, EquiAV  | 0.5  |
> | Self-supervised  | AudioCLIP, ImageBind, Wav2CLIP  | 0.32 |
>
> Table 2: #Params vs $\rho$
> | Model | #Params | $\rho$|
> |----------|----------|----------|
> | UAVM |199M | 0.42  |
> | CAV-MAE  | 191M  | 0.54 |
> | EquiAV | 173M  | 0.55 |
> | AudioCLIP  | 134M  | 0.41 |
> | ImageBind  | 1.2B  | 0.36 |
> | Wav2CLIP  | 314M  | 0.21 |
>
> Table 3: Architecture vs $\rho$
> |  | Models | $\rho$|
> |----------|----------|----------|
> | ResNet | AudioCLIP, Wav2CLIP  | 0.31  |
> | Transformers  | UAVM, CAV-MAE, EquiAV, ImageBind  | 0.46 |
>
> Table 4: Augmentation vs $\rho$
> | Augmentations? | Models | $\rho$|
> |----------|----------|----------|
> | Yes | EquiAV, AudioCLIP, ImageBind  | 0.44  |
> | No  | UAVM, CAV-MAE, Wav2CLIP | 0.39 |
>
> 4. Comparison with single-modality corruptions: Thank you for the experimental suggestion! We repeat the same inference experiment but with unimodal corruptions (Tables 5,6). Please find the single-modality TTA results in Tables 7 and 8 of our submission. In Tables 5 and 6 below, we add our proposed audio/visual corruptions to the test sets of AudioSet, VGGSound, and Kinetics-Sounds.
>
>
> Table 5: Results on the test set of datasets with audio corruptions at a severity level of 5. Metrics are averaged across 15 tasks. Performance drops, relative to clean test, are in brackets.
>
> | Model     | AS mMAP ↑ | VGG mAcc ↑ | KS mAcc ↑  |
> |-----------|-----------|------------|------------       |
> | CAV‑MAE   | 38.19(-11.67)     | 50.59(-14.91)      | 74.47(-3.65)      |
> | EquiAV    | –         | 51.81(-10.09)      | 77.63(-8.39)      |
> | AudioCLIP | 19.31(-9.67)     | 16.97(-9.81)      | 41.60(-9.41)      |
> | ImageBind | 13.93(-4.42)     | 10.69(-17.49)      | 18.62(-33.84)      |
>
>
> Table 6: Results on the test set of datasets with visual corruptions at a severity level of 5. Metrics are averaged across 15 tasks. Performance drops, relative to clean test, are in brackets.
>
> | Model     | AS mMAP ↑ | VGG mAcc ↑ | KS mAcc ↑  |
> |-----------|-----------|------------|------------        |
> | CAV‑MAE   | 46.27(-3.59)     | 56.18(-9.32)      | 73.13(-4.99)      |
> | EquiAV    | –         | 51.49(-10.41)      | 74.16(-11.86)     |
> | AudioCLIP | 23.06(-5.93)     | 24.57(-2.21)      | 31.97(-19.04)      |
> | ImageBind | 15.86(-2.49)     | 27.68(-0.5)      | 50.56(-1.9)      |
>
> We observe that, under unimodal corruptions, models are more robust comparatively. Models seem to prefer audio for inference. However, improving the robustness on bimodal shifts (very likely in the real-world) is still an open question.
>
>
>
>
> [1] Hendrycks, D., Dietterich, T.: Benchmarking neural network robustness to common corruptions and perturbations. arXiv preprint arXiv:1903.12261 (2019)

---

> > ### Comment · Reviewer_5mV3 · 2025-08-05
> >
> > The authors have addressed my concerns, and I have no further questions.

---

> > > ### Author Response · Authors · 2025-08-07
> > >
> > > Thank you for taking the time out to review our work and rebuttal!

---

### Decision · Program_Chairs · 2025-09-18

**Decision:**

Accept (poster)

**Comment:**

This paper introduces AVROBUSTBENCH, a benchmark evaluating audio-visual model robustness against correlated, co-occurring corruptions across both modalities. Its core claims are: (a) Existing models degrade significantly under increasing bimodal corruption severity; (b) Current test-time adaptation (TTA) methods fail in such scenarios; (c) The proposed AV2C method—adapting QKV attention via low-entropy samples—improves robustness. Strengths include the novelty of correlated bimodal corruption modeling (addressing real-world shifts), comprehensive coverage (75 corruptions across 4 datasets), and high utility (open-sourced datasets/code). During rebuttal, authors resolved concerns. While questions on corruption realism remain, the rebuttal adequately addressed methodological critiques. The work’s impact lies in its pioneering focus on bimodal robustness, rigorous evaluation scale, and actionable resources. Given its alignment with DB Track’s emphasis on data-centric AI benchmarks and resolved core issues, acceptance is recommended.